# Instruments of Mineral Deposit Safeguarding in Poland, Slovakia and Czechia—Comparative Analysis

**Slávka Gałaś [1,*](ID), Alicja Kot-Niewiadomska [2](ID), Andrzej Gałaś [2](ID), Julián Kondela [3](ID) and Blažena Wertichová [4](ID)**

1 Faculty of Geology, Geophysics and Environmental Protection, AGH University of Science and Technology, al. Mickiewicza 30, 30-059 Krakow, Poland

2 Mineral and Energy Economy Research Institute, Polish Academy of Science, Provision of Mineral Policy, Wybickiego 7A, 31-261 Krakow, Poland; a.kn@min-pan.krakow.pl (A.K.-N.); agalas@min-pan.krakow.pl (A.G.)

3 Institute of Geosciences, Technical University of Košice, Letná 9, 040 01 Košice, Slovakia; julian.kondela@tuke.sk

4 Faculty of Mining and Geology, VSB-Technical University of Ostrava, 17. listopadu 2172/15, 70800 Ostrava-Poruba, Czech Republic; blazena.hamadova@vsb.cz

* Correspondence: sgalas@agh.edu.pl

**Abstract:** Mineral deposits are essential for the economic, technological and social development. However, to enable them to play an appropriate role in the process of sustainable development, they need to be safeguarded in a comprehensive and systemic manner in the same measure as other elements of the environment. The practice of securing access to areas where the mineral deposits can be found is based on the statement that they can be extracted only in places where they occur. This fact defines the type and scope of instruments for safeguarding prospective deposit areas of minerals and their documented deposits. These issues gained in significance in the EU level in recent years however views on this subject across the Member States still vary. The paper subjects instruments of mineral deposit safeguarding used in Poland, Slovakia and Czechia to the analysis and multi-criteria comparative assessment. It recommends their division into the conceptual, legal, spatial planning and economic ones. As a result of studies, similarities and differences in the approach to mineral deposit safeguarding in individual countries are shown, indicating good practices and suggesting possible changes. The analysis revealed many analogies in actions aimed at mineral deposit safeguarding in individual countries, however the assessment of their effectiveness and implementation points at the necessity of taking further steps to increase deposit safeguarding.

**Keywords:** instruments of mineral deposit safeguarding; comparative analysis; mineral deposits





## 1. Introduction

The main environmental challenges faced by the European Union have evolved since first days of the European environmental policy, originated at the European Council meeting in Paris in 1972. EU has quickly become a world leader in this respect presenting one of the most stringent regulatory standards [1]. In the 1970s and 1980s, however, the focus was put on such traditional environmental issues as the protection of species and the improvement of air and water quality as well as reduction of emissions [2]. Nowadays, stress is placed on a more systematic approach taking into account the relations between various issues and their global aspect, while the array of elements under protection is larger [3]. The environment is a set of natural elements, including those transformed as a result of human activity, in particular the following: surface of the earth, minerals, waters, air, landscape, climate and other elements of biodiversity as well as mutual relations among these elements [3,4]. Their protection sets down natural conditions of organisms' existence, man included and is fundamental to their further development [3,5–7].

Following this view, legal enactments of multiple European countries, Poland, Slovakia and the Czech Republic included, laying down—in general terms—rules for the

environmental protection, also pronounce that mineral deposits are its integral part and hence they should be covered by comprehensive safeguarding under separate provisions [4,8,9]. Additionally, mineral deposits are characterised by non-renewability as well as a specified and unchangeable location resulting from the geological structure, which makes them significantly different from other components of the environment. At the same time, growing human population and increased living standards raise the demand and affect the scarcity of natural resources, including mineral raw materials. Therefore, the list of challenges that Europe should address trying to retain its existing position includes, among other things, the risk of depletion of non-renewable natural resources vital for the contemporary development and the resultant rise in their prices [10,11].

Constantly growing demand for minerals generated by dynamic and multi-directional development of the global economy makes it necessary to use minerals rationally and effectively, search for new solutions aimed at reducing the dependence on raw materials and, first of all, protect their resources successfully. However, a constantly growing pressure for the development of spatial planning directions other than mining is a factor that intensifies the necessity to take steps toward their effective safeguarding, as these directions often tend to be given priority. The most important ones are the development of urban, agricultural and tourism areas, the latter requiring high environmental quality standards, landscape being part of it. In accordance with the traditional understanding of mining, it is a part of economy causing degradation of the environment [12] forgetting that current requirements for the mines with respect to environmental protection are very stringent. There is low awareness of the fact that in the result of correct reclamation of post-mining sites attractive places for tourists, beneficial for the general public, as well as natural habitats protecting valuable species of plant and animals may be created [13,14]. Therefore it is not a problem of quantity of resources but access to them as well as possibilities of their use determined by economic, environmental and social factors become the primary issue e.g., [15–17] and the *sine qua non* condition of their safeguarding. In this context the primary issue for a successfully rationale and sustainable supply of minerals to society is granting the access to them in spatial planning [18]. Taking into account the formal-legal conditions of analysed countries—mineral deposits safeguarding is a multidimensional topic and includes the following elements:

1. Safeguarding of land access and safeguarding of area against development, which may prevent the use of the deposit and the necessary mining activities;
2. Resources safeguarding against unjustified losses and minimising unavoidable losses (also resulting from inappropriate land development);
3. Complete use of documented resources of deposit including accompanying mineral.

Thus, the deposit safeguarding includes both the safeguarding to access to them and their rational and efficient management during mining operations. In this sense, it is analysed in present article.

Mineral deposit safeguarding should be based on the precise knowledge on the specific country's resource base [19,20] which will allow to create a far-sighted concept of their safeguarding and use and then to apply it in practice by means of varied tools. It should be mentioned that the mineral deposit safeguarding needs to be implemented both in the context of prospective areas of their occurrence as well as undeveloped and existing deposits. Depending on the status of deposit development, instruments of its safeguarding will vary, yet they should always result in the rational management of raw materials and comprehensive use of minerals, including accompanying minerals.

Existing studies presenting the subject of mineral deposit safeguarding in Poland have been executed primarily on the national level [21–31], while the inter-state comparative analysis has appeared in single publications and related only to Slovakia [31]. Multiple times, the studies focused on only one of many safeguarding tools [32–36] or a specific group of minerals [37,38], without the comprehensive analysis of the issue of deposit safeguarding. Moreover, in Slovakia and Czechia generally this topic has not been discussed in academic papers. This is because of the general belief that deposit safeguarding is

satisfactory. There is, however, a constant need of searching for innovative, more efficient tools adopted to current and future circumstances. The aim of this paper is conducting multidimensional assessment of instruments used in Poland, Slovakia and Czechia for safeguarding forecasted areas of mineral occurrence and their documented deposits. This is the first scientific paper considering the above issue that was developed in an international consortium. At the same time, identification of the best, most effective examples of management instruments for rational use and safeguarding of deposits is the essential subject of the study.

## 2. Material and Methods

This paper analyses and evaluates the existing instruments of mineral deposit safeguarding in Poland (PL), Slovakia (SK) and Czechia (CZ) and a division into conceptual, legal, spatial planning and economic ones has been proposed (Figure 1). It should be mentioned that the study focuses on forecasted areas of mineral deposit occurrence, undeveloped deposits and exploited deposits.

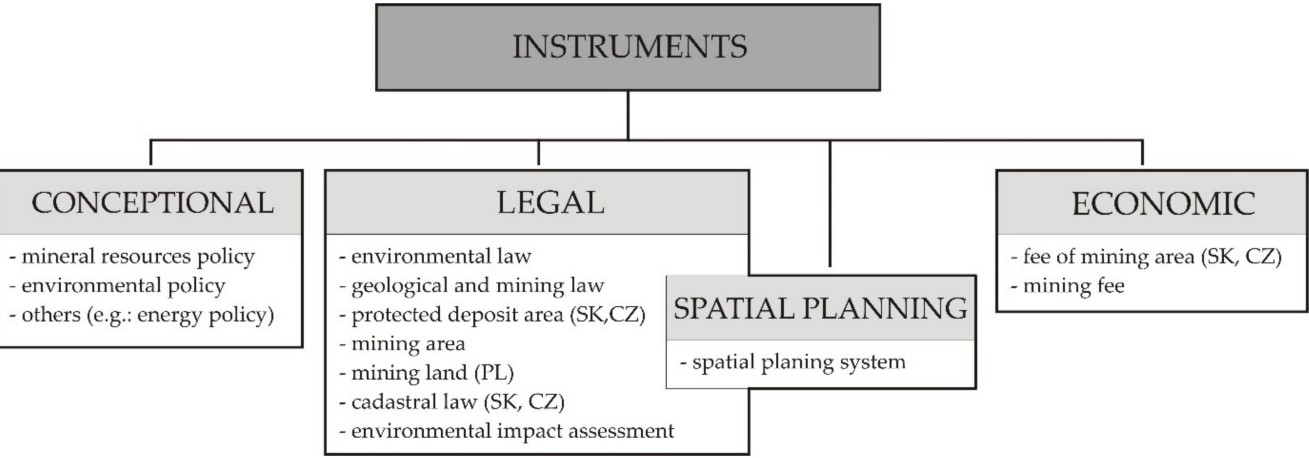

**Figure 1.** Classification of the analysed instruments of mineral deposit safeguarding (PL—Poland, SK—Slovakia, CZ—the Czech Republic).

Description of the instruments has been preceded by an analysis of mineral deposit safeguarding on the level of the European Union, a general description of raw material characteristics in individual countries (taking into account available sources of data on their raw material potential) and an overview of definitions and mineral classifications in individual countries.

Further, according to the suggested classification, instruments of mineral deposit safeguarding have been characterised. The conceptual instruments were considered broadly defined, long-term and comprehensively directed documents describing the current status and prospects for a country's development in specific area as well as laying down basic goals and tools for their implementation. In the context of deposit safeguarding, they comprise such strategic documents of individual countries as, among other, the mineral policy and the ecological (environmental) policy (Figure 1). The mineral policy should be a set of overarching principles and actions for reducing the risk in the supply of raw materials aimed at securing multiannual economic and social needs of a country resulting from priorities adopted for the economic development. As the mineral policy implemented by a specific country directly affects the environment, society and the economy, it should be included in the sustainable development policy as its integral part.

Legal instruments include legal norms (statutes) in force on the territory of a given country which, in a direct or indirect way, govern the matter of mineral deposit safeguarding in analysed countries. As spatial planning constitutes its undisputed foundation,

the authors decided to set it apart from the group of legal instruments and analyse on a separate basis (Figure 1).

Economic instruments are first of all aimed at influencing the behaviour of entities (deposit users in this case) and providing incentives for their environment-friendly behaviours. The state, through its economic instruments which significantly affect effective management of environmental resources, forces constructive activities beneficial for the environment. As they are characterised by indirect coercion as their main feature, those instruments constitute a paramount tool used in ecological policy [39]. Moreover, their role is to collect financial means, which enables providing financial support for pro-environmental projects.

All of the aforementioned instruments have been analysed and evaluated in terms of their scope, functionality, mode of action, suitability and effectiveness for deposit safeguarding. The comparative method with a stress on institutional and legal aspects has been used.

A primary source of the assessment were legal regulations, strategic documents, planning documents with the practice of their implementation as well as own experiences and observations made for individual countries. As a result of studies conducted in such a manner, similarities and differences in the approach to mineral deposit safeguarding in individual countries have been shown, indicating good practices and offering possible suggestions/conclusions in the aspect of mineral deposit safeguarding.

## 3. Results

### 3.1. The Subject of Mineral Deposit Safeguarding on the EU Level—Outline

The issue of mineral deposit safeguarding as an indispensable element of the environment in Poland, Slovakia and Czechia is consistent with constitutional principles e.g., [40,41] and lies in the framework of national systemic solutions in individual countries. The policy of mineral safeguarding as national natural resources does not lie, however, within competences of the European Union but solely within competences of its Member States at the national or regional level, less often at the local level [42,43]. Nevertheless, there have been numerous discussions on whether the mineral deposit policy at the European level should exist and, if so, which issues should it address within the framework of the European sustainable industrial strategy [20,44,45]. On the one hand, it is believed that such a policy would surpass the powers of the Commission, but on the other hand, some Member States could see real value in establishing a pan-European policy stance [20]. Moreover, it might allow for gradual actions in the area of social education not only at the local level, but also international level [44]. Mineral deposits would be then taken equally to other natural resources, in the case of which the European Union is equipped in numerous competencies (e.g., Directive on the conservation of natural habitats and of wild fauna and flora, Directive establishing a framework for Community action in the field of water policy, Directive on ambient air quality and cleaner air for Europe).

Despite all this, directions of activities in the area of mineral deposit safeguarding in individual countries are set by EU actions taken within the framework of the Community's mineral policy. Their starting point is the so-called Raw Materials Initiative of 2008 [43] in which it is stated that the 'access to mineral resources and their affordability are a factor decisive for proper functioning of the European Union economy'. The document have then set the strategy concerning access to mineral resources in which it is the second pillar that promotes the need to safeguard mineral deposits—each country is supposed to identify and safeguard access to significant mineral deposits in accordance with its national legislation [43].

In order to counteract negative trends in the development, including the decrease of European Union's economic competitiveness, in the next step the EC in 2010 adopted a document titled 'Europe 2020: A Strategy for Smart, Sustainable and Inclusive Growth' [11]. The strategy covers three inter-related priorities:

1.  Smart growth—economic growth based on knowledge and innovation;
2.  Sustainable growth—supporting the economy that uses resources more effectively, is more environmental-friendly and more competitive;
3.  Growth fostering social inclusion.

Securing sustainable access to raw materials is vital for the competitiveness and growth of the European economy and for the goals laid down in the Strategy Europe 2020 [11].

In 2011, the European Commission drafted a document titled 'Tackling the challenges in commodity markets and on raw materials' [10] which, for the first time, defined raw materials critical for the European Union and described the EU commercial strategy in non-energy raw materials. The document, i.e., presented directions for more effective management of resources and defined such future directions for the Raw Materials Initiative as supporting supplies from internal sources of the European Union and supporting effective management of raw material resources. Thus, the importance of three EU raw material security pillars defined in the document of 2008 has been sustained [46].

In September 2013 the European Innovation Partnership on raw materials (EIP RM), operating under the auspices of the EC, announced the Strategic Implementation Plan for this Partnership (SIP) as an implementation of the Raw Materials Initiative. Most important tasks which may concern the mineral deposit safeguarding are as follows [47]:

1.  Coordination of research and development studies in the scope of mineral resources;
2.  More effective exploration;
3.  Innovative techniques of extraction, treatment and processing of minerals,
4.  Development of mineral resources substitution,
5.  Improvement of legal regulations on conducting mining operations and waste management,
6.  Introduction of a framework for national mineral policies,
7.  Development of knowledge on mineral resources.

Another issue, however, directly related to the EU mineral policy, is identification of raw materials critical for the economy of the Community. The first list of raw materials critical for the European Union was prepared in 2011 as one of the priority actions taken with the framework of the aforementioned Raw Materials Initiative [43]. Since that moment, it has been updated several times and the number of critical raw materials has expanded from 14 in 2011 to 20 in 2014, 27 in 2017 and 30 in 2020 [48–50]. It confirms the European Union's growing demand for raw materials accompanied by increased limitations on their availability, including primary sources located in the EU Member States.

The above actions of the EU clearly indicate on the growing awareness of the Community that providing reliable, fair (consistent with preserving generational justice) and sustainable supplies of raw materials (including mineral resources) is important to maintaining its industrial base which is the key element of growth, prosperity and competitiveness of the EU [51]. As it is stressed by the European Commission and working groups, mineral deposits' safeguarding must be secured through the properly implemented spatial policy, but it must be based on the following elements: knowledge of resources, transparent methodology of their identification, long-term forecasts of the demand for raw materials and proper identification of resources that require safeguarding [20,45].

It is worth mentioning that MINATURA 2020 and MinLand projects, financed by the European Commission through the Horizon 2020 programme, have also targeted the issue of mineral deposit safeguarding and the proper integration of mineral policy with spatial policy at the European, national and regional level. The MININATURA 2020 programme initially intended to create an outline of the so-called Raw Material Directive which would regulate the matter of deposit safeguarding at the EU level. The project was however concluded solely by general recommendations in this scope [51,52] in line with the demands made by the European Commission. Preparing the detailed and comprehensive guidelines turned out to be problematic because of substantial differences in the identification of the resource base and in implementation of mineral and spatial policy in individual EU countries. These issues have also been continuously addressed in the

MinLand project which currently provides a very good database on the raw material policy and spatial planning of most of the EU countries. Recommendations for their integration are primarily based on properly selected case studies presenting good practices in land usage on every stage of mining operations (from research to reclamation) [53].

The countries analysed in this paper made an attempt to integrate the provisions of aforementioned EU documents through national documents implementing the mineral policy of a given country. Their detailed description will be provided in Sections 3.2 and 3.3.1.

*3.2. General Raw Material Characteristics of Analysed Countries*

3.2.1. Definition and Classification of Minerals and Their Deposits

In Poland, a mineral deposit is a natural aggregation of mineral, rock and other, material extraction of which may bring economic benefit. Deposits defined in such a way are divided into: those covered by mineral right propered to the State Treasury and covered by the ownership right to the land property [54] (Table 1).

In Slovakia and in Czechia, minerals are defined as solid, liquid and gaseous part of the earth's crust which are divided into exclusive (reserved) minerals and non-exclusive (non-reserved) minerals (Table 1). Reserved minerals are minerals important for industry and energy generation or minerals of high value. According to the mining law in force [55,56], the reserved mineral deposits form a mineral wealth property of the state. Deposits of non-reserved minerals constitute a part of a land property, which practically means that such deposits are the property of a land owner and are not covered by legal protection under the mining law [55–57].

Waters, except for therapeutic, thermal and brine waters are not considered minerals in Poland, while in Slovakia and Czechia exceptions include waters and natural therapeutic waters (subject to some exceptions—see Table 1), medicinal muds and peats [54–56].

3.2.2. Raw Material Potential and Access to Data

Raw material potential of Poland is substantial and, at the same time, diversified. The level of identification of the domestic resource base can be described as good. According to data as of 31.12.2018 in total 14,532 mineral deposits have been identified in Poland. Amongst them 44 are deposits of metallic minerals, 59 of chemical minerals, 719 of energy minerals (including methane from coal seams) and the most numerous groups of rock minerals of 13,575 deposits. In total, 5264 deposits are in permanent or periodical exploitation (Table 2) [58]. The deposits are unevenly dispersed across the country. Their highest concentration can be found in southern (including Lower Silesian, Silesian and Lesser Poland provinces) and central (including Holy Cross and Łódź provinces) Poland. Altogether, over 500 million tonnes of minerals are mined in Poland [58], of which approximately 68% are rock minerals (mainly sands and gravels as well as construction and road stones), 24% are energy minerals (mostly bituminous coal and lignite) and 6.5% are metal ores, mainly Cu-Ag. For many years Poland has been one of European leaders in extraction of copper ores, silver, bituminous coal and lignite.

According to data as of 31.12.2018, in Slovakia 1160 mineral deposits were registered in total. Amongst them 47 are deposits of metallic minerals, 20 are deposits of industrial ores, including chemical ones, 97 are energy minerals and the most numerous group of rock minerals represented by 277 deposits (Table 2). Of them, 104 deposits are mined, the majority of which are deposits of rock minerals. The deposits are unevenly dispersed across the country with the highest concentration in the central part of Slovakia. In total, over 40 million tonnes of minerals are mined in the country, of which 53% are rock minerals (mainly limestone, dolomite, magnesite and limestone clay), 5% are energy minerals (mainly brown coal and lignite), 41% are industrial minerals and the remaining 1% are metal (gold) ores [59]. In recent years showed the growing interest in magnesite deposits and possible mining of metallic magnesium from magnesites and dolomites as well as mineral deposits for the construction industry. For future purposes, the presence of selected critical raw materials in the historically exploited deposit of Ni-Co ores is verified. Mercury

and molybdenum ores are also expected to become useful, in case of which verification of documented balance resources is carried out. Slovakia is the richest in carbonate minerals raw materials compared to the surrounding countries.

**Table 1.** Mineral deposits classification applied in Poland, Slovakia and Czechia (based on: [54–56]).

| Mineral Deposits | Poland | Slovakia, Czechia |
|---|---|---|
| Deposits covered by mineral right propered to the State Treasury (PL)/Reserved deposits (SK, CZ) | - Hydrocarbons,<br>- Hard coal,<br>- Methane existing as an associated mineral,<br>- Lignite,<br>- Metal ores excluding bog iron ores,<br>- Native metals,<br>- Radioactive ores,<br>- Native sulphur,<br>- Rock salt, sylvinite,<br>- Potassium-magnesium salts,<br>- Gypsum and anhydrite,<br>- Precious stones,<br>- Rare-earth elements,<br>- Noble gases, wherever they occur<br>- Curative water, thermal water and brine | - Radioactive minerals,<br>- All kinds of coal, oil and natural gas, bituminous rocks for energy use,<br>- Minerals for industrial metal production,<br>- Magnesite,<br>- Minerals for industrial production of phosphorus, sulphur and fluorine,<br>- Rock salt, potassium, boron, bromine and iodine salts,<br>- Graphite, barite, asbestos, mica, talc, diatomite, glass and foundry sand, mineral pigments, bentonite,<br>- Minerals for industrial production of REE and semiconductor elements,<br>- Granite, granodiorite, diorite, gabbro, diabase, serpentinite, dolomite and limestone, if they are polishable and mineable in blocks, travertine,<br>- Crystals and gemstones for technical use,<br>- Halloysite, kaolin, ceramic and refractory clays and claystones, gypsum, anhydrite, feldspar, perlite and zeolite,<br>- Quartz, quartzite, limestone, dolomite, marl, calcareous clays (only CZ), basalt, clinkstone, trachyte if they are suitable for chemical and technological processing and smelting,<br>- Mineralised waters for reserved minerals production,<br>- Other natural gases for technical use |
| Deposits covered by the ownership right to the land property (PL)/Deposits of non-reserved minerals (SK, CZ) | Deposits of minerals not listed above (e.g., natural aggregates, dimension and crushed stones, dolomites, limestone, kaolin, ceramic clays, glass sands, feldspar raw materials), | Deposits of minerals not listed above (sands, gravels, building stone, brick raw materials) |
| NOT minerals (*natural resources that are not defined as mineral raw materials and for which there are no documented deposits*) | Water (excluding curative water, thermal water and brine) | - Waters, excluding mineral waters from which minerals may be extracted industrially<br>- Natural healing waters and natural mineral table waters from which minerals can be extracted industrially<br>- Medicinal mud and other products of natural medicinal resources,<br>- Peat, clay, sand, gravel and pebbles in riverbeds, provided that they do not contain minerals in mineable quantities,<br>- Cultural layer of soil which is vegetative environment of plants (only CZ) |

**Table 2.** Basic information on the resource base of Poland, Slovakia and the Czech Republic, as of December 31, 2018 (based on [58–60]).

| | General Information | | | Resources Basis [Number of Deposits] * | | | | | | | | | | | |
|---|---|---|---|---|---|---|---|---|---|---|---|---|---|---|---|
| | Area [km$^2$] | People [mln] | Total extraction [mln t] | Metallic | | | Industrial/Chemical ** | | | Energetic | | | Rocky *** | | |
| | | | | e | ue | total | e | ue | total | e | ue | total | e | ue | total |
| Poland | 312 696 | 37.97 | 511 | 9 | 35 | 44 | 12 | 47 | 59 | 373 | 346 | 719 | 4 792 | 8 782 | 13 575 |
| Slovakia | 49 035 | 5.458 | 41.1 | 2 | 45 | 47 | 3 | 17 | 20 | 27 | 70 | 97 | 72 | 205 | 277 |
| Czech Republic | 78 866 | 10.69 | 126 | 0 | 64 | 64 | 96 | 350 | 446 | 113 | 141 | 254 | 495 | 1091 | 1586 |

e—exploited deposits, ue—unexploited deposits, * without deposits of water, **only chemical in PL, *** only construction minerals (i.e., brick raw materials, crushed and dimension stone, sand and gravel) in CZ.

According to data as of 31.12.2018, in Czechia totally 2355 mineral deposits were registered [60]. Amongst them 64 are deposits of metallic minerals, 446 are deposits of industrial ores, including chemical ones, 25 are energy minerals and the most numerous group of rock minerals represented by 1586 deposits (Table 2). Of them, 704 deposits are mined, the majority of which are deposits of rock minerals (Table 2). The deposits are unevenly dispersed across the country with the highest concentration in the northern and eastern part of Czechia. In total, 126 million tonnes of minerals are mined in the country, of which 51% are rock minerals (mainly crushed stone and sand and gravels, but also brick minerals and dimension stone), 34% are energy minerals (nearly all of them is lignite) and the remaining 15% are industrial minerals (i.e., kaolin, bentonite, clays, glass sands, feldspars, diatomite, dolomite, limestone and in lower amounts also precious stones and gypsum). Despite the large resource base of metal ores documented (incl. lithium, manganese, tin, tungsten and gold ores) the Czech Republic has not been mining those deposits since 2017 (this is when the last uranium mine was closed) [60].

Referring to the mineral deposits classification (presented in Table 1) in Poland 957 deposits (6.5%) are covered by the mineral right (including 135 deposits of underground waters classified as minerals) and the remaining ones are covered by the ownership right to the land property (Table 3). Amongst the deposits covered by the mineral right 472 are exploited (approx. 50%; most of them are fossil fuels) and more than 4.7 thousands of rock mineral deposits from the other group. Nearly 350 million tonnes of rock minerals are extracted from deposits covered by the ownership right to the land property. The production of minerals from deposits covered by the mineral right amounted to over 162.5 million tonnes—total of solid energy minerals and crude oil, metal ores and chemical minerals and additionally over 5 billion m$^3$ of gases (including mainly natural gas).

**Table 3.** Mine production of minerals from deposits covered by the mineral right (PL)/reserved deposits (SK, CZ) and deposits covered by the ownership right to the land property (PL)/deposits of non-reserved minerals (SK, CZ) (according to [58–60]) (as at the end of 2018).

| | Deposits Covered by the Mineral Right (PL)/Reserved Deposits (SK, CZ) | | | Deposits Covered by the Ownership Right to the Land Property (PL)/Deposits of Non-Reserved Minerals (SK, CZ) | | |
|---|---|---|---|---|---|---|
| | PL | SK | CZ | PL | SK | CZ |
| Number of deposits | 957 | 642 | 1582 | 13,575 | 518 | 773 |
| Protected deposit areas (PDA) | no | 519 | 1147 | no | no | no |
| Mining areas | 472 | 450 | 960 | 4792 | no | no |
| Number of mined deposits | 472 | 201 | 536 | 4792 | 121 | 168 |
| Mining (kt) | 162,577 (kt) 5.2 mld m$^3$ | 30,871 | 114,000 | 349,060 | 10,196 | 12,000 |

In Slovakia and in Czechia, it is possible to observe different proportions in breakdown of deposits and volume of extraction in particular groups of raw materials (Table 3). In Slovakia, amongst the registered 1160 mineral deposits more than 55% represent reserved mineral deposits. For 519 of them a protected deposit area (PDA, details in Section 3.2.2) has been delimited and for 450 of them also a mining area. In Czechia (as at the end of 2018), over 67% of all registered mineral deposits are the reserved deposits. For 1147 of them a protected deposit area has been delineated and for 960 of them a mining area. In Slovakia and in Czechia, currently 31% of reserved deposits are under exploitation with the extraction of over 30 million tonnes and over 114 million tonnes respectively (Table 3). In the case of non-reserved deposits in Slovakia and in Czechia, more than 22% are exploited with the extraction of over 10 million tonnes (SK) and over 12 million tonnes (CZ) (Table 3).

The main source of information about the documented deposits in Poland is their geological documentation kept at the National Geological Archives. It is the basis for the creation of descriptive data collected in the System of Management and Protection of Mineral Resources in Poland [61]. The most important information on the deposits of mineral resources in Poland, their state of management as well as the volume of extraction can be found in the yearbook 'The Balance of Mineral Resources in Poland' [58] and on the website Mineral Resources of Poland [62]. They are prepared on the basis of reports sent by entrepreneurs for the previous year. They are supplemented by the register of mining areas in Poland [63]. The information is completed by spatial data: boundaries of deposits as well as mining areas and mining lands delineated for exploited deposits. They are free of charge and publicly available through the Central Geological Database [64]. Collecting and making the geological information available is an obligation of the Polish Geological Survey whose function since 2012 has been fulfilled by the Polish Geological Institute–National Research Institute.

The aggregate list of reserved deposits and the balance of mineral resources in Slovakia is kept by the Ministry of Environment. The balance is updated every year on the basis of reports sent by the entrepreneurs for the previous year. The balance also contains information on the type of a mineral deposit, name of the mining area owner, list of minerals according to their degree of identification and selected parameters of reserved deposits. Additionally, every year the Ministry of Environment prepares the evidence of non-reserved mineral deposits of the Republic of Slovakia. A set of spatial information concerning the geological and mining data used, among other things, to issue opinions on investment projects and the planning documentation can be found on the websites of Geofond which is run by the National Geological Institute of Dionýz Štúr. In the nearest future, a system of making available the vector data will be launched [65].

The method of registering deposits and mineral resources is laid down in the mining law [66] and it is performed by the Czech Geological Survey (CGS) on behalf of the Ministry of Environment or the Ministry of Industry and Trade. The information is collected from annual reports of mining organisations and takes the form of a list in the 'Balance of reserved resources of minerals' and the 'Register of non-reserved deposits of minerals' [67]. The data contain information on the condition, changes and movement of resources or the value of extraction in the previous year. These reports are not available to the public but a part of the information is reflected in the collective publication of the Czech Geological Survey—'Mineral Commodity Summaries' [68] annually published in Czech and English languages and available online on the CGS website. The spatial data on registered deposits, safeguarded deposit areas, mining and research areas as well as data from related registers is available in the 'Raw Materials Information System–SurIS' [69].

*3.3. Instruments of Mineral Deposit Safeguarding*

3.3.1. Conceptual Instruments

Mineral Policy

Poland still does not have the mineral policy, even though the domestic geological community has commenced efforts for the development of such a document over 80 years

ago [70]. Establishing in 2016 a plenipotentiary of the government and an inter-services team for the mineral policy has become a milestone in achieving this goal. In 2018, the draft version of the National Mineral Policy objectives was sent for public consultations and approved a year later [71]. This document, for the first time, takes into account the interests of raw material security of Poland in such a comprehensive and holistic manner. The goal of the mineral policy is the rational management of mineral resources and setting directions for investments in this field in accordance with the current state-of-the-art and stage of development which contributes strengthening of the country in the international arena [72]. The NMP draft points at the necessity of safeguarding and sustainable management of the mineral deposit resources, in particular of strategic importance for the domestic and regional economy in the context of the system of spatial planning [71]. Identification of strategic minerals requires, however, a comprehensive valorisation of deposits which will become a basis for their categorisation and division into deposits of supranational (European), national and regional importance [73]. Unfortunately, throughout the previous year, legislative work over the document was ceased and it still has a draft status.

The mineral policy of the Republic of Slovakia in the field of mineral resources was approved by the government in 1995. The document is mainly focused on the use of domestic mineral resources due to the long-term economic and social development needs in relation to the environmental aspects of sustainable development, beginning from the research and identification of mineral deposits to their mining and safeguarding. The mineral policy concerns only deposits of reserved minerals. In the current amendment of the mineral policy (of 2004) deposits strategic for the economy and depleted minerals have been defined [74]. Stipulations of the national mineral policy are further developed in detail at the level of regional mineral policies. Their main goal is, among other things, the determination of space and time limitations in the mining of mineral deposits within spatial development plans of local administrative units in relation to the land volume/absorption. They should mandatorily specify, in the set time, the scope and sequence of mining to be carried out, when it will be concluded and when reclamation work will be performed [74].

The new raw material policy for minerals and their resources is the country's primary conceptual tool for the management of minerals and their deposits, and their safeguarding is one of its paramount goals [75]. It was adopted by the Ministry of Industry and Trade of the Czech Republic (MIT) in 2017. Stipulations of this document are a binding criterion of evaluation and decision-making for national administrative authorities. In Czechia, similarly to Slovakia, the mineral policy refers only to reserved deposits of minerals. Main objectives of the policy is to provide sufficient supplies of raw materials to the Czech Republic, their efficient use in accordance with the needs of economic and social development with particular focus on the environmental protection and sustainable development. Deposit safeguarding should be closely followed and ensured for both mined and unmined deposits—as a potential source of a specific mineral in the future—and deposits of strategic and critical raw materials currently unexploited for various reasons (e.g., selected deposits of lignite, bituminous coal, uranium). The mineral policy of the Czech Republic also pays attention to the necessity of designating backup spots for minerals close to depletion and for regions where their consumption is expected to rise [75]. After the raw materials policy was issued in 2017, Czechia adopted the report on the need of securing national economic interests in using critical superstrategic raw materials of the European Union and some other raw materials [76]. The main purpose of this and other related documents is to strengthen the role of the state in exploration and extraction of superstrategic minerals (the list of European critical raw materials refined according to the needs of the CZ). This document envisages, among other things, the evaluation and identification of deposits of strategic importance for the Czech Republic and, consequently, to consider further measures for their safeguarding and possible future use under the state control [76].

Other Conceptual Instruments

One of the foundations of the environment protection policy in Poland is the National Environmental Policy 2030 (2019) which discusses also the issues of the geological resources management, pointing at the need to develop and implement the country's mineral policy. It justifies this need by the fact that the rational management of geological resources is a condition of long-term economic security of the country, and thus its national security. At the same time, it identifies a potential threat related to mineral deposits. According to the document, the most important one is the use of areas located directly above mineral deposits. In particular, it refers to deposits located in areas of rapid urbanisation, in territories covered by the protection of other natural and landscape resources or relevant to other strategic interests of a country [77].

In Poland, also the Strategy 'Energy Security and Environment–perspective until 2020' (2014) is worth noticing, according to which it is of key importance to continue the identification of energy minerals' location and create opportunities for their exploitation within the country's territory as well as to identify the strategic deposits. This will ensure safeguarding of such deposits against activities which might prevent their mining and will allow designating the area solely for purposes related to its identification and exploitation. Such protection should also cover deposits whose exploitation at the moment (for various reasons) is not viable [78].

The Environmental Policy Strategy of the Slovakian Republic until 2030 was adopted in 2019. It pointed at the need of conducting a wide spectrum of geological work in order to ensure the sustainable development of the society and safeguarding of the rock environment with necessary coordination of geological environment potential and threats as well as geological risk. For strategic raw materials and investments, also vital economic and social aspects are in place which should be taken into account in order to use the deposits in a rational manner [79].

The National Environmental Policy of the Czech Republic 2012–2020 (2011, amendment 2016) addresses the issue of mineral deposits only marginally. Mostly in connection to the effective use of mineral resources, it highlights an importance to minimise the impact on the environment during and after their extraction. The document e.g., mentions the need to use indigenous mineral resources and increase self-dependence of the Czech Republic in the area of energy raw materials [80].

Other documents of domestic importance that may impact the mineral raw materials management are: Energy Policy of Poland (2009), Slovakia (2014) and the Czech Republic (2015), Politics of Secondary Raw Materials of the Czech Republic (2019) as well as the National Strategy for Sustainable Development of Slovakia (2001) [81–85]. Among the conceptual instruments for mineral deposit safeguarding one should mention the National Spatial Development Concept in Poland (2011), the Conception of Territorial Development of Slovakia (2011) and the Spatial Development Policy in the Czech Republic (2020) [86–88] which are discussed in detail in Section 3.3.3—Instruments of Spatial Planning.

### 3.3.2. Legal Instruments

The issue of mineral deposit safeguarding in Poland is dispersed in at least four legislative acts. They include, first of all, the Acts: on the environmental protection [89], geological and mining law [54], on preserving the national character of strategic natural resources of the country [90] and on spatial planning and development [91]. The latest of these is described in detail in the section entitled: Instruments of Spatial Planning. Apart from the aforementioned acts, selected legal provisions concerning the issue of deposit management and mineral deposit safeguarding can be found in several statutes and related implementing regulations. However, they are analysed in this paper.

The Act of 27 April 2001 the Environmental Protection Law [89], first of all, defines the rules for safeguarding mined deposits. Deposit mining is executed in an economically justified manner, using measures that limit the environmental damages, while ensuring rational extraction and the comprehensive management of minerals (including accompa-

nying minerals). The operator commencing or executing the mining is obliged to take steps necessary to safeguard deposit resources. The manner of safeguarding the unmined deposits proposed in this legislative act is taking into account the areas of their occurrence as well as present and future exploitation needs in local (at the commune level) planning documents. These issues are further elaborated in the Act of 9 June 2011 the Geological and Mining Law [54] and the Act of 27 March 2003 on Spatial Planning and Development [91].

The most important Polish legislative act regulating the mining activity and specifying requirements for deposit safeguarding is the Geological and Mining Law (further: GML) [54]. According to the GML: 'documented mineral deposits (...) shall, for protection purposes, be disclosed in studies of conditions and directions for the spatial development of a commune, local spatial development plans and province spatial development plans'. Within 2 years from the geological report (or 6 months for hydrocarbon deposits) being approved by the competent geological administration authority, the area of a documented deposit must be entered into the study of conditions and directions of spatial development of a commune. The costs of drawing up an amendment to the planning documents shall be borne in full by the commune or entrepreneur but only in the case of drawing up documentation for hydrocarbon deposits. If the commune authorities fail to observe the time limit referred to in the Act, the authority introducing the deposit to the study is a province governor.

The Geological and Mining Law also specifies rules and conditions of extraction of minerals from deposits, thus ensuring the safeguarding of mined deposits. For safe operation of the mining entity and the rational management of a deposit it is also obligatory that the communal planning documents additionally take into account the mining area (MA) and the mining land (ML) for the deposit mined (Figure 2). The designation of MA and ML takes place in the concession decision on the request of an entrepreneur [54]. The MA defines space in which the entrepreneur is authorised to conduct mining operations on the rules specified in the concession, while the ML defines space which may be adversely affected by the mining operations. Therefore, it is possible to properly plan the use of lands and locations of new sites in a manner that will minimise the possible adverse effect on neighbouring lands and, at the same time, ensure full access to the resources through the proper management of the land surface above the deposit and within its direct neighbourhood.

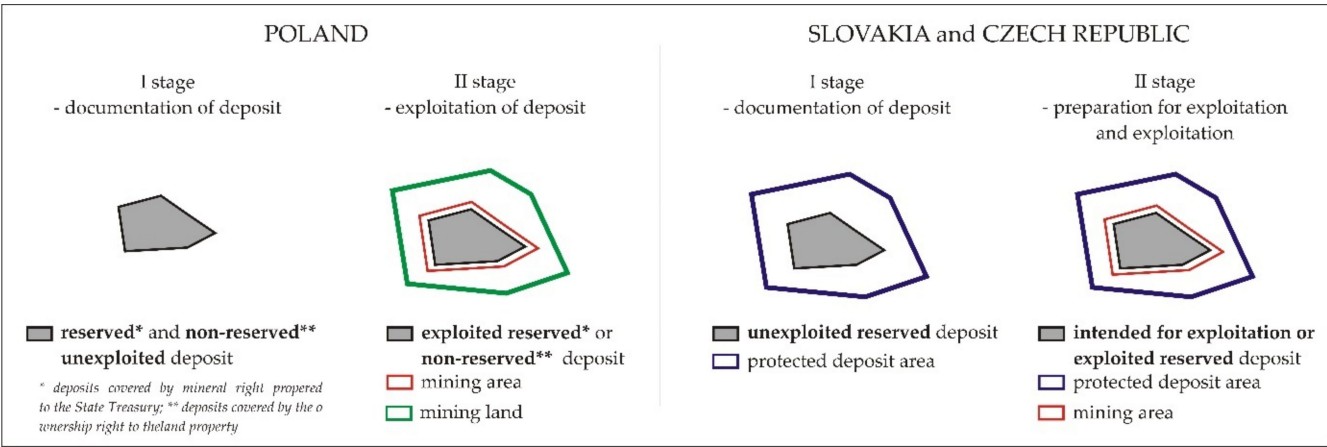

**Figure 2.** General scheme of determined of protected deposit area, mining area and mining land (based on [54–56]) [own study].

The Act of 6 July 2001 on Preserving the National Character of Strategic Natural Resources of the Country [90] specifies that the strategic natural resources of the country include for example deposits covered by mineral right propered to the State Treasury (listed in detail below). The use of such deposits must be conducted in accordance with the principle of sustainable development and for public benefit.

Legal regulations on mineral deposit safeguarding in Slovakia and Czechia originate in the period of common legal order, when these countries formed the Socialist Republic of Czechoslovakia (year 1988). After its dissolution in 1993 into two politically independent states—the Republic of Slovakia and the Czech Republic—both legal systems have been substantially transformed and today are characterised by the sound foundation but different approaches to some issues.

Legal frameworks of Slovakian and Czech mining consist of the following legislative acts: The Act on the Protection and Use of Mineral Resources (commonly known as Mining Act) [55,56], the Act on Mining Activities, Explosives and State Mining Administration (commonly known as Mining Activity Law) [92,93] and the Act on Geological Work (commonly known as Geological Act) [94,95].

Legislation on the mineral deposit safeguarding in both countries is concentrated in the Mining Law [55,56] and is also comprised in the Act on Town and Country Planning and Building Code [96,97], whose provisions in this matters are laid down in detail in the section titled 'Instruments of Spatial Planning'.

The common purpose of mining and geological law in both countries is to ensure mineral deposit safeguarding. The aforementioned division of minerals into the reserved and non-reserved minerals is of key importance for their legal status, ownership rights and legal regime for the safeguarding and use of these minerals [55,56]. The Mining Act [55,56] applies to safeguarding of the reserved deposits for which the Ministry of Environment issued a reserved deposit certificate. In this case, two safeguarding instruments are used for a reserved deposit: protected deposit area (PDA) and mining area (MA) (Figure 2) [55,56,98].

The primary legal form for safeguarding of reserved deposits in Slovakia and Czechia is the protected deposit area which provides protection of a reserved deposit against circumstances that might threaten its future use for commercial purposes, i.e., its extraction. Boundaries of PDA are larger than boundaries of a reserved deposits as they cover an area on which buildings and equipment not connected with the extraction of a deposit and could prevent or obstruct extraction, which in turn might pose a threat to the adjacent objects. PDA specifies conditions for deposit safeguarding by defining limitations in the use of lands, which need to be taken into account in town and rural spatial planning and designating areas for fulfilling specific social and economic roles [99,100].

The procedure of PDA defining is initiated upon the request of an organisation or a national administration authority, and in Slovakia additionally by a commune. The application to delimit a protected deposit area in Slovakia is submitted within 3 months from serving a certificate on documenting a reserved deposit (reserved deposit certificate). The decision on determining PDA in Czechia is entrusted to the Ministry of Environment in cooperation with the Ministry of the Industry and Trade (MIT) and the District Mining Authority. In Slovakia, only the Mining Authority is authorised in this matter. These institutions discuss the issue with the spatial planning authority and the building authority and, in Slovakia, additionally with the authority competent for nature conservation. MIT and the District Mining Authority promote in proceeding the reserved deposit safeguarding and benefits of its commercial use. At the same time, spatial planning authorities, the construction authority and the environmental protection authority take into account and promote the interests of land use and environmental protection that might contradict PDA delimitation [55,56,99,100].

According to the mining law in Slovakia and Czechia, the spatial planning authorities and authorities drawing up spatial planning documentation are obliged to conduct their activities based on the documentation of forecasted deposit areas and reserved mineral deposits provided to them by the Ministry of Environment. Those authorities must suggest solutions which are the best possible from the standpoint of safeguarding and use of mineral resources and other legally protected general interests. Moreover, the law imposes an obligation to base the activity of spatial planning authorities on the outcomes of geological work [55,56]. As a result, in both countries boundaries of a protected deposit

area must be entered into the spatial planning documentation and the land and building register [55,56,101,102]. Documents of PDA are kept by the Ministry of Environment and their public access is very limited, covering only such basic information as the location, name, identification number and type of a mineral for which protection has been determined [65,69]. Information on the date of establishing the PDA, the applicant and limitations of land use determined is not made available. According to the cadastre and mining law in Slovakia, the updated register of protected deposit areas is kept by the Real Estate Cadastre Information System [103,104].

A decision on determining PDA is a mandatory appendix to the application concerning the determination of a mining area determined for the extraction of a reserved deposit of the specific mineral or the group of minerals. The reserved deposits should be rationally extracted, i.e., as completely as possible and with minimum losses and pollution, taking into account the current technical and economic conditions and with the elimination of unnecessary negative impacts on the occupational and environmental safety. Determination of MA is carried out on the basis of the existing protected deposit area taking into account the possible extraction of neighbouring deposits as well as the impact of extraction on the environment. The mining area is delimited by a competent District Mining Authority in cooperation with State Administration Authorities (environmental protection, spatial planning, construction office). A decision on delimitation and alteration of the mining area is a land purpose decision and, therefore, boundaries of the mining area delimited in such a manner must be entered to the spatial planning documentation pursuant to the Building Act [55,56,96,97].

A condition for delimitation of a mining area is also the settlement of any conflicts of interest, if any, or if the settlement was not reached during delimitation of the PDA. The conflicts of interests may pertain to the objects and/or interests of natural or legal persons, protected under special provisions of law, which may be threatened by the extraction of a reserved deposit. If this is the case, approaches that would enable the use of a reserved deposit while ensuring necessary protection of such objects and interests must be suggested. To this end, the parties must sign an agreement which would determine whether a threatened object or interest should be safeguarded, to what extent, or for which period. If such an agreement is not reached or if the general economic interest of utilising such a reserved deposit outweighs the justified interests of a property and other real estate owners (in the opinion of a competent ministry), in Slovakia there is a possibility to acquire a property or right to property through a decision on expropriation or through the acquisition of user's rights. Until today, only the instrument of acquisition of user's rights for the state (for 30 or 50 years) has been used, e.g., for the magnesite deposit in Košice. In the Czech Republic, a decision of settling the conflict of interest is in the first instance taken by regional authority. An agreement between parties is valid one month after its application if the regional authority does not disagree. In the case, an agreement was not accomplished or the regional authority disagrees, the Ministry of Industry and Trade after a consultation with the Ministry of Environment and the Mining Authority of the Czech Republic, in cooperation with other competent national administration authorities, taking into account the opinion of regional authorities, is the one who decides [55,56].

The necessity of safeguarding of minerals is also indicated in Cadastral Laws of Slovakia and Czechia [101,102] and, additionally, in Act on Environmental Impact Assessment of all three countries analysed [105–107]. The real estate cadastre, as an information system in Slovakia and Czechia, is used in particular for the protection of rights to real estates, for taxes and levies, for property evaluation, for conservation of agricultural and forestry areas, for the protection of minerals as well as protected areas and natural formations. The correct PDA and MA registration in the land and building register may significantly contribute to the safeguarding of mineral deposits due to the fact that the real estate cadastre and the information it contains are the basis for constructing other systems on real estate information and spatial planning [101,102].

Pursuant to the provisions of the Environmental Impact Assessment Acts [105–107], it is also required to carry out an assessment of the impact on mineral resources for projects

of policies, strategies or planning documents (the so-called strategic assessment) and investments (e.g. road construction). It is aimed to eliminate the limitations in the access to deposits. Even though the process of environmental impact assessment varies in individual countries, for example in the scope of screening, documentation and requirements for persons drawing up environmental studies, its basis and goal are shared: it is the protection of the environment, including its resources, in view of construction work or redevelopment of undertakings planned [26,28].

### 3.3.3. Instruments of Spatial Planning

In Poland, according to the Act of 23 March 2003 on Spatial Planning and Development [91], a hierarchical system of spatial planning is in force for which planning documents different in scope and character need to be drawn up. Thus, the spatial policy is implemented through the following documents (Table 4):

- National spatial development concept (national level, further: NSDC);
- Province spatial development plans (regional level, further: PSDP);
- Studies of conditions and directions of spatial development of a communes (SCDSD), local spatial development plan of a commune (LSDP) (local level).

**Table 4.** The most important elements of spatial planning systems in Poland, Slovakia and Czechia, related to mineral deposit safeguarding [91,96,97].

| Level | Poland | | | Slovakia | | Czech Republic | | |
|---|---|---|---|---|---|---|---|---|
| | Accompanying Documents | Basic Documents | Spatial Planning Materials | Spatial Planning Documentation | Spatial Decision | Spatial Planning Materials | Spatial planning Documentation | Spatial Decision |
| National | Set of analytic reports | National Spatial Development Concept | Urban studies, Territorial General, Territorial prognosis, Territorial technical documents | Conception of Territorial Development | - | Planning analytical materials, Planning study | Spatial Development Policy | - |
| Regional | Ecophysio-graphic study | Province Spatial Development Plan | | Territorial plan of the region | Decision on land use Decision on a protected part of the country * | | Development Principles | Decision On Alteration Of The Use Of An Area * |
| Lokal | | Study of Conditions and Directions of Spatial Development of a commune, Local Spatial Development Plan of a commune | | Territorial plan of the village | | | Land use Plan Regulation Plan | |

\* If applicable to more than one village unit, decision is issued at regional level.

The above documents are supplemented by various studies, among which the following should be mentioned: ecophysiographic study, i.e., documentation prepared for the purposes of land-use plans at the regional and local level, characterising individual natural elements, including mineral deposits in the area covered by SCDSD or LSDP, and their mutual linkages [89], including the negative impact of intended activities specified in the mining concession on the environment [54] (Table 4).

The primary aspect of mineral deposit safeguarding in the National Spatial Development Concept 2030 (2011) is an obligation to draw up the so-called mineral deposit exploitation plans which would ensure deposit extraction in accordance with their value [86]. Their preparation will enable the management of a land property and of deposits connected and unconnected with the land, reducing the social conflict potential and at the same time safeguarding the state's strategic interests and interests of the local communities. An additional tool of deposit safeguarding proposed by NSDC is entering, on the basis of the said exploitation plans, all potential concession areas in planning documentation of all levels. Such areas would be covered by a spatial planning provision within a specified time horizon laid down in the integrated strategy of province development in consultation with the minister competent for economy and environmental matters. Additionally, pursuant to NSDC, it is necessary to draw up a list of minerals strategic for Polish economy and

include them in the planning documents. The document points directly at the safeguarding of strategic deposits of energy minerals, highlighting at the same time that even if for social, economic or other reasons like certain known deposits of energy minerals remain unused, they should be treated as a permanent special strategic resource that must be protected by special legal measures against different kinds of human activity, in particular investments (incl. housing, construction of transport infrastructure). It refers to bituminous coal and lignite deposits. Until today, however, no official work has been conducted over the premises of this document [33].

In Poland, stipulations of the higher level planning documents should be taken into account in documents of the lover level. The primary instrument of the spatial policy, however, are the local spatial development plans, drawn up and adopted only at the commune level. Only these documents as local legislative acts set forth regimes of investment activity within an area. Paradoxically, however, their enactment is not mandatory [91]. As a result (acc. to the status as of the end of 2018), only 30.8% of the country was covered by LSDPs. This value, nevertheless, has been in an upward trend over the recent years [108]. In the absence of LSDP, taking up and conducting the mining activity is permissible only when it will not violate the manner of property use set forth in the study of conditions and directions of spatial development of a commune and in separate provisions [54]. This matter in particularly important for the procedure of obtaining the mining concession in Poland. The application for granting a concession should therefore include information on the purpose of land in the boundaries of which the activity will be conducted, specified especially in the local spatial development plan and in separate provisions. If the LSDP or (eventually) the SCDSD do not envisage mining activity within the specific area, obtaining the exploitation rights must be proceeded by the procedure of land conversion. According to the Act on Spatial Planning and Development [91], consistently with the Environmental Protection Law [89] and in line with the Geological and Mining Law [54], it is mandatory for the planning documentation to take into account the conditions resulting from the occurrence of documented mineral deposits and mining lands. Theoretically, every documented deposit should then be taken into account in the aforementioned documents and the manner of land development should not prevent and/or restrict the management of a deposit, which is stipulated in detail in the Geological and Mining Law [54]. Numerous cases, however, indicate that this provision is not correctly implemented [21,29,31,109].

The spatial planning documents currently in force in Slovakia and Czechia originate in the Act No. 50/1976 Town and Country Planning and Building Code (referred to as the Building Act) [96] originating in the period of the Socialist Republic of Czechoslovakia and in respective implementing regulations [110,111]. In Slovakia, this law (with numerous amendments) is in force until today. In Czechia, spatial planning is currently regulated by the Act No. 183/2006 on Town and Country Planning and Building Regulations (Building Act) [97].

According to the Building Act, instruments of spatial planning in Slovakia consist of: studies for spatial planning (spatial planning materials), spatial planning documentations and spatial decisions (Table 3) [96]. The primary tool of territorial development and care for the environment of the country, regions and village units is spatial planning documentations. All other sectoral concepts pertaining to the economic, social or cultural development must comply with the spatial planning documentations in force. The documentations are drawn up on the basis of land-use planning elaborations (such as the study, the forecast, spatial and technical documents) and consist of: the conception of territorial development, the territorial plan of the region, the territorial plan of the village unit and of the zone village unit [96] (Table 4). The spatial planning documentations are subject to agreement with relevant mining institutions: conception of territorial development with the Higher Mining Office and territorial plans of regions, village units and zones of village units with a District Mining Office. Planning documentations in the scope of safeguarding and use of mineral deposits are also assessed by the Ministry of Environment [55].

When drawing up spatial development plans of regions and village units, spatial-technical documents are of paramount importance, creating a set of data regarding the potential of a specific region, in particular on the status of use of natural resources, development of individual residential elements, settlement of conflicts of interests and types of socio-economic activity. Drawing up a spatial development plan of a region and a village unit also includes defining and delimiting mineral prospection areas, safeguarding deposit areas and mining areas as well as relates to both the consequences and the technical support of forecasted extraction of a mineral [110].

In Slovakia, erection of buildings, conversion of land and protection of important interests within a specific territory are possible only on the basis of spatial decisions, divided into: decision on the location of construction, decision on the land use, decision on protected area or protected zone and decision on excluding an area from development. The spatial decisions are made and issued based on spatial development plans of village units/zones or spatial-technical documents if such spatial development plans of village units have not been drawn up. A mining area for a specific deposit is established through the decision on the land use, while a protected deposit area is created by the decision on protected area [55,56,96].

In Czechia, until 2006 instruments of spatial planning similar to the Slovakian instruments were in force. Only the amendment of the Building Act introduced such new instruments as spatial analytic documents and a regulation plan (Table 4) [97,111].

The spatial analytic documents are drawn up to the scale appropriate for regions and village units as well as verify possibilities and conditions for land use. These documents are relevant to safeguarding mineral resources as they assess the condition and development of an area, determining the so-called limits of land use, i.e., restrictions in land use due to the protection of general interests, including mineral deposits (forecasted reserved deposits, reserved deposits and non-reserved deposits), protected deposit areas and mining areas. The spatial planning materials constitute, together with the spatial studies, the basis for drawing up planning documents: development principles of regions and land use plans of village units. The development principles of regions are a foundation for, among other things, amending the spatial development policy which is a binding document for preparing and issuing the rules of spatial development, regulation plans and spatial decisions. Detailed conditions for the use of land, location and spatial arrangement of buildings may be set forth in the regulation plan [97,111].

In Czechia, the spatial decisions are decisions on the location of a building or equipment, conversion of land use, conversion of the environmental impact of a building, division or merger of land and on the protective zone. Delimitation of a mining area is a decision on the conversion of land use [97]. Contrary to Slovakia, establishing of a PDA is not a spatial decision.

In Czechia, the Ministry of Environment, Ministry of Industry and Trade as well as District Mining Authorities issue opinions to spatial development plans and regulation plans in the scope of safeguarding and use of mineral resources [56].

### 3.3.4. Economic Instruments

In Poland, economic instruments related to safeguarding of exploited mineral deposits are introduced by the Geological and Mining Law [54]. An entrepreneur that has obtained a concession for extraction of a mineral from the deposit and, in the case of concessions for prospecting and exploration of hydrocarbon deposits as well as extraction of hydrocarbons from a deposit has obtained an investment decision, pays the mining fee determined as a product of its rate and the quantity of mineral extracted from the balance and off-balance deposit, in the settlement period. The rates of mining fees for individual types of minerals are set forth in the Appendix to the aforementioned law [54] (Table 5). The rates of fees are subject to annual changes in correspondence with the average annual consumer price index for total consumer goods and services, planned in the budget act for a given calendar year. The mining fee does not consider market prices of the mineral extracted. To activity

conducted with gross violation of conditions provided for in a concession or in an approved geological work plan or in a geological work plan subject to notification, an additional charge will apply. The additional charge for the extraction of minerals is determined in the five-fold rate of the mining fee set for a given type of mineral, multiplied by the quantity of the mineral extracted in this way. An increased charge will also apply to activity conducted without a required concession or without an approved geological works plan or without a geological works plan subject to notification. The increased charge for the extraction of minerals is determined in the forty-fold amount of the mining fee set for a given type of mineral multiplied by the quantity of the mineral extracted without the concession.

**Table 5.** Economic instruments for deposit safeguarding in Poland (PL), Slovakia (SK) and Czechia (CZ); (based on: [54,112,113]).

| Economic Instruments | PL | SK | CZ |
|---|---|---|---|
| The mining area fee (SK, CZ) | - | 665 EUR/km$^2$/year | Unexploited mining area 11 EUR/ha/year * Exploited mining areas 36.4 EUR/ha/year * |
| The mining fee (PL)/The fee for extracted mineral (SK, CZ) | Based on the rates of mining fees for individual types of minerals, they are subject to annual valorisation | Calculation of payments is based on the share of extraction costs and total costs of manufacturing of products | Based on the rates of mining fees for individual types of minerals |

\* rates in CZ converted into EUR, exchange rate as at: 10/30/2020, EUR 1 = CZK 27.250 [114].

In the field of mineral deposit safeguarding, Slovakian and Czech mining laws provide for two types of fees. They are: a fee from the mining area and a fee for the mineral extracted (corresponding with the royalty binding in Poland) (Table 5) [55,56].

A basis of the fee from the mining area is its surface given in square kilometre (SK) or in hectares (CZ), in accordance with its delimitation on the land surface. In Slovakia, an entity owning a delimited mining area, when the Government of the Republic of Slovakia has not announced the program of restricting the extraction of a reserved deposit in the mining area, pays a fee in the amount of EUR 665 annually per each square kilometre of the mining area [55]. In Czechia, the rate of fee is CZK 300 per hectare of a delimited mining area on which the extraction is not performed, and CZK 1000 if the extraction activity consisting preparation, opening and extraction in the mining area has been permitted (Table 5) [56]. Unexploited mining areas are treated in the privileged manner due to the lack of adverse effect on the environment. The payment term in both countries is a calendar year. In Poland, a fee for a delimited mining area does not apply.

In Slovakia and in Czechia, a payer of a fee for extracted minerals is a holder of the mining area, i.e., an enterprise conducting exploitation on a given area on the basis of a valid mining concession; and an enterprise that obtained extracted minerals during deposit prospection within a delineated prospecting area, and an enterprise that prospects or extracts a deposit without permission [55,56]. In the Czech Republic, the fee consists of a sum of partial fees determined for each type of minerals extracted, and the rate of a payment is determined separately for each type of a mineral, pursuant to governmental regulations. The maximum amount of the fee is limited to 10% of market price per quantity unit of a given type of a mineral. In the case of lignite extracted using open-pit methods, a partial basis for payments is the product of the quantity of the mineral extracted and its quality, i.e., calorific value. The payment term for a fee for minerals extracted is a calendar year [56,112].

In Slovakia, the fee for minerals extracted is calculated per quarter of a calendar year as the product of three elements: (1) The ratio of the cost of mineral extraction to the total cost of manufacturing products from the mineral extracted; (2) revenues from the sale of products from the mineral extracted; and (3) a rate of a fee for particular types of a mineral. The maximum amount of the fee for extracted minerals does not exceed 20% of a market price for products manufactured from extracted radioactive minerals and 10% of a market price for products manufactured from other extracted minerals [55,113].

## 4. Discussion of the Results

*4.1. Comparative Analysis of Mineral Deposits Classification and Access to Data*

Poland, Slovakia and Czechia have a substantially varied, both in terms of quality and quantity, base of raw materials. On the one hand it results from the size of these countries, but on the other hand on their raw material potential directly related to their geological structure and a degree of its identification. All three countries are characterised by many centuries of mining traditions.

A substantial difference may be noticed between the analysed countries in the assignment of minerals to deposits covered by the mineral right (PL) and corresponding reserved deposits (SK, CZ) which in all the countries belong to the State Treasury as well as mineral deposits covered by the ownership right to the land property (PL) and non-reserved deposits (SK, CZ) (Table 1). In Slovakia and in Czechia, apart from deposits of fossil fuels, metals and chemicals, a number of rock mineral deposits are considered as reserved deposits which, in Poland, are usually covered by the ownership right to the land property (such as dimension stones, glass sands, kaolin). In Czechia and in Slovakia such classification results in the existence of legally standardised tools of their safeguarding (PDA, MA). In Poland, however, covering deposits by the mineral right does not guarantee their safeguarding by appropriate tools. Deposits of lignite and Zn-Pb ores may be the best example as, despite considering them strategic raw materials, they have been for many years the subject of social conflicts and numerous administrative proceedings making their management impossible. The status of therapeutic waters, thermal waters and brines is also extremely different. In Poland, they are covered by the mineral right of the State Treasury, while in Slovakia and in Czechia they are classified as underground waters and not considered minerals (Table 1) [54–56].

As it was mentioned at the beginning, the basis of successful safeguarding of deposits should be their comprehensive and available database. In this context, all of the countries analysed need to be positively assessed for easy and generally available access to the geological and mining information. A wide scope of data on the documented, exploited and unexploited deposits is made available free of charge through databases, which is in every case regulated by respective legislative acts. In Poland and in Czechia, it also refers to vector data necessary to perform spatial analyses, including those related to deposit safeguarding. A comment may be made only to the information on PDAs in Slovakia and in Czechia. In these countries, for example data on restrictions in the land use resulting from delimiting a protected area are not generally available.

*4.2. Comparative Analysis of Mineral Deposit Safeguarding Instruments*

A detailed comparative analysis of identified instruments of mineral deposit safeguarding in Poland, Slovakia and Czechia needs to be proceeded by defining authorities at the national administration level competent in the field of deposit safeguarding. In this scope, certain similarities must be noted. Two ministries are responsible for issues related to safeguarding mineral deposits in various stages of their development. They are: the Ministry of Environment and the Ministry of State Assets, whose counterpart in Slovakia is the Ministry of Economy and, in the Czech Republic, the Ministry of Industry and Trade. Deposit safeguarding, mineral policy in the case of Slovakia and Poland and awarding of concession lie within the competences of the Ministry of Environment (in Czechia the Ministry of Industry and Trade).

### 4.2.1. Conceptual Instruments

The assessment of conceptual instruments in individual countries showed that the most important instrument, regulating the issue of mineral deposit safeguarding, should be a properly defined mineral policy of a country which is an overarching document determining the correct directions and the detailed tools. In this matter, among the countries analysed, circumstances in Poland are the least favourable. Even though work in this scope has been continued for nearly several dozen years, the valid document is still missing

(Table 6). In this paper, only the draft document has been assessed over which the work was carried out in the years 2016–2018. The project of Mineral Policy [71] in its main objectives demonstrates great similarity to analogical documents in force in Slovakia and in Czechia. Every time, one of main goals of the mineral policy is safeguarding of mineral deposits compliant with sustainable development principles and closely integrated with the spatial planning. The assessment of conceptual documents shows that mineral policies of all three countries point at the vital issue of identifying deposits strategic for the country's development. The documents obviously differ in the scope of solutions offered which result directly from internal conditions of individual countries. In Poland, the draft of mineral policy refers also to forecasted areas which is missing in Slovakia and in Czechia. It should be also noted that in Slovakia and in Czechia the mineral policy is further expanded and continued in regional raw material policies. Additionally, in Slovakia and in Czechia it refers solely to the reserved deposits, i.e., those which are owned by the State. In Poland, it pertains to both groups of documented deposits.

The up-to-date character of strategic documents is also very important (e.g., the last amendment to the Mineral Policy in Slovakia was adopted in 2004) as they should include, among other things, needs for the consumption of mineral raw materials and their prices, changing with time and location. A part of the documents assessed, such as the National Environmental Policy of the Czech Republic 2012–2020 (2011) [80] and the Energy Policy of Poland (2009) [81], currently is in the process of amending.

In line with the above considerations, it the assessment of implementation of strategic documents' provisions related to deposit safeguarding, in the scale 1–3 (1—unsatisfactory or below satisfactory, 2—satisfactory, 3—effective or very effective) only the Mineral Policies of Slovakia [74] and the Raw Material Policy of Czech Republic [75] as well as the Spatial Development Policy of the Czech Republic [88] obtained the grade 'satisfactory', while the remaining documents have been assessed as unsatisfactory or below satisfactory (Table 6).

### 4.2.2. Legal Instruments

Many similarities between the analysed countries can be found in legal tools used for mineral deposit safeguarding. Every time, their starting point is the mining law [54–56] which regulates the rules and conditions for establishing, conducting and concluding of operations consisting in mineral extraction. These laws point at the necessity of mineral deposit safeguarding in national systems of spatial planning, which is subsequently addressed in respective regulations [91,96,97]. The implementation of these provisions in Poland has been assessed as satisfactory, but their efficiency a grade lower—unsatisfactory— particularly for unexploited deposits. In Slovakia and in Czechia, the implementation of safeguarding and its efficiency have been assessed as satisfactory (Table 7). Deposit safeguarding, precisely defined in the mining laws of Slovakia and Czechia [55,56], obtained positive assessment in the view of the topic analysed in this paper, with a reservation that legal safeguarding refers solely to reserved deposits (Table 7). It consists in establishing a protected deposit area which defines restrictions in the use of land due to the presence of a reserved deposit (Figures 2 and 3). The effectiveness of so defined safeguarding of reserved deposits obtained a satisfactory grade (Table 7). The scope of restrictions may vary for individual PDAs, however it always affects ownership rights to the property, in particular to lands and buildings. Land development for other purposes than mining in protected deposit areas are not permitted if they would limit access to the deposit. Such other forms of activity as farming, forestry or tourism may still be conducted. A decision on establishing the PDA affects also the prices of land on which the PDA has been established or of its neighbourhood [115,116]. In Poland, no form of deposit safeguarding which would correspond to PDA and fulfil its functions can be found. Therefore, the assessment points at the lack of defined tools for the enforcement of deposit safeguarding, and the effectiveness of deposit safeguarding in Poland has been considered unsatisfactory (Table 7).

**Table 6.** Assessment of selected conceptual instruments in terms of directions for defined activities and of their implementation in comprehensive mineral deposit safeguarding in Poland (PL), Slovakia (SK) and Czechia (CZ); Explanation: Y: yes, N: not, -: not applicable; Assessment of implementation of strategic documents' provisions related to deposit safeguarding in the scale 1–3 (1—unsatisfactory or below satisfactory, 2—satisfactory, 3—effective or very effective) [own study].

| Country | Conceptual Instruments | Directions of Activities | | | | | | | | | |
|---|---|---|---|---|---|---|---|---|---|---|---|
| | | Mineral Deposit Safeguarding as a One of Goal of Documents (Y/N) | Safeguarding of Forecasted Areas (Y/N) | Safeguarding of Mineral Deposit Covered by Mineral Right Propered to the State Treasury/Reserved Deposit (Y/N) | Safeguarding of Mineral Deposit Covered by the Ownership Right to the Land Property/Deposits of Non-Reserved Minerals (Y/N) | Safeguarding of Unexploited Deposits (Y/N) | Safeguarding of Exploited Deposits (Y/N) | Selecting Strategic Deposits and/or Defining the Method of Their Determination (Y/N) | Topicality of Documents (Y/N) | Integration with Spatial Planning (Y/N) | Assessment of the Implementation of Safeguarding Regulations (1–3) |
| PL | The National Mineral Policy * [71] | Y | Y | Y | Y | Y | Y | Y | Y | Y | 1 |
| | The National Environmental Policy 2030 (2019) [77] | Y | N | Y | Y | Y | Y | Y | Y | Y | 1 |
| | The Energy Policy of Poland until 2030 (2009) [81] | Y | N | Y | Y | Y | Y | Y | N | Y | 1 |
| | The National Spatial Development Concept 2030 (2011) [86] | Y | N | Y | Y | Y | N | Y | Y | Y | 1 |
| SK | The Mineral Policy of the Republic of Slovakia in the field of mineral resources (amendment 2004) [74] | Y | N | Y | N | Y ** | Y ** | Y ** | Y | Y | 2 |
| | The Environmental Policy Strategy of the Slovakian Republic until 2030 (2019) [79] | Y | N | Y | Y | N | Y ** | Y ** | N | Y | 1 |
| | The Energy Policy of Slovakia (2014) [82] | Y | N | Y | N | N | Y ** | Y ** | N | Y | 1 |
| | The Conception of Territorial Development of Slovakia (2011) [87] | Y | N | Y | N | Y ** | Y ** | N | Y | N | 2 |
| CZ | Raw Material Policy of the Czech Republic in the Field of Mineral Resources and their Resources *** (2017) [75] | Y | N | Y | N | Y ** | Y ** | N ** | Y | Y | 2 |
| | The National Environmental Policy of the Czech Republic 2012–2020 (2011) [80] | N | N | N | N | N | N | N | Y | Y | - |
| | The State Energy Policy of the Czech Republic (2015) [83] | N | N | N | N | N | N | N | Y | N | - |
| | The Spatial Development Policy of the Czech Republic (2020) [88] | Y | N | Y | N | Y ** | Y ** | N | Y | Y | 2 |

* document not approved, ** only applies to reserved deposits of minerals, *** in 2017 was adopted the Report on the need of securing national economic interests in using critical superstrategic raw materials of the European Union and some other raw materials [76].

**Table 7.** Comparison and assessment of the most important aspects of legal instruments (including instruments of spatial planning) for mineral deposit safeguarding in Poland (PL), Slovakia (SK) and Czechia (CZ); explanation: Y: yes, N: not; -: not applicable; assessment of the implementation of provisions and of effectiveness of deposit safeguarding in the scale 1–3 (1—unsatisfactory or below satisfactory, 2—satisfactory, 3—effective or very effective) [own study].

| Legal Instruments | | | Elements to Be Assessed | | | | | | | | | | | | | | | | | | |
| --- | --- | --- | --- | --- | --- | --- | --- | --- | --- | --- | --- | --- | --- | --- | --- | --- | --- | --- | --- | --- |
| | | | Safeguarding of Forecasted Areas (Y/N) | | | Safeguarding of Reserved Deposits/Deposits of non-Reserved Minerals (Y/N/Y/N) | | | Safeguarding of Exploited/Unexploited Deposits (Y/N/Y/N) | | | Defining Enforcement Tools for the Deposit Safeguarding (Y/N) | | | Assessment of the Implementation of Regulations (1–3) | | | Assessment of Safeguarding Effectiveness (1–3) | | |
| Poland | Slovakia | Czechia | PL | SK | CZ | PL | SK | CZ | PL | SK * | CZ * | PL | SK * | CZ * | PL | SK | CZ | PL | SK * | CZ * |
| The Geological and Mining Law (2011) [54] | The Act on the Protection and Use of Mineral Resources (1988) [55] | The Act on the Protection and Use of Mineral Resources (1988) [56] | N | N | Y ** | Y/Y | Y/N | Y/N | Y/Y | Y/Y | Y/Y | N | Y | Y | 2 | 2 | 2 | 1 | 2 | 2 |
| The Environmental Protection Law (2001) [89] | Nature and Landscape Protection Act (2002) [8] | Nature and Landscape Protection Act (1992) [9] | N | N | N | Y/Y | N | N | Y/Y | N | N | N | N | N | 2 | - | - | 1 | - | - |
| The Act on spatial planning and development (2003) [91] | The Act on Town and Country Planning and Building Code (1976) [96] | The Act on Town and Country Planning and Building Code (2006) [97] | N | N | Y ** | Y/Y | Y/N | Y/N | N/Y | Y/N | Y/Y | N | N | N | 1 | 2 | 2 | 1 | 1 | 1 |
| The Geodetic and Cartographic Law (1989) [117] | The Cadastral Law (1995) [101] | The Cadastral Law (2013) [102] | N | N | N | N/N | Y/N | Y/N | N/N | Y/N | Y/Y | N | N | N | - | 2 | 2 | - | 1 | 1 |
| The Act on Sharing Information about the Environment and its Protection, Public Participation in Environmental Protection and Environmental Impact Assessment (2008) [105] | Act On Environmental Impact Assessment (2006) [106] | Act On Environmental Impact Assessment (2001) [107] | N | Y ** | Y ** | Y/Y | Y/N | Y/N | Y/Y | Y/Y | Y/Y | N | N | N | 1 | 1 | 1 | 1 | 1 | 1 |

* only applies to reserved deposits of minerals, ** their application in the Land Use Plan, without any guarantee of safeguarding.

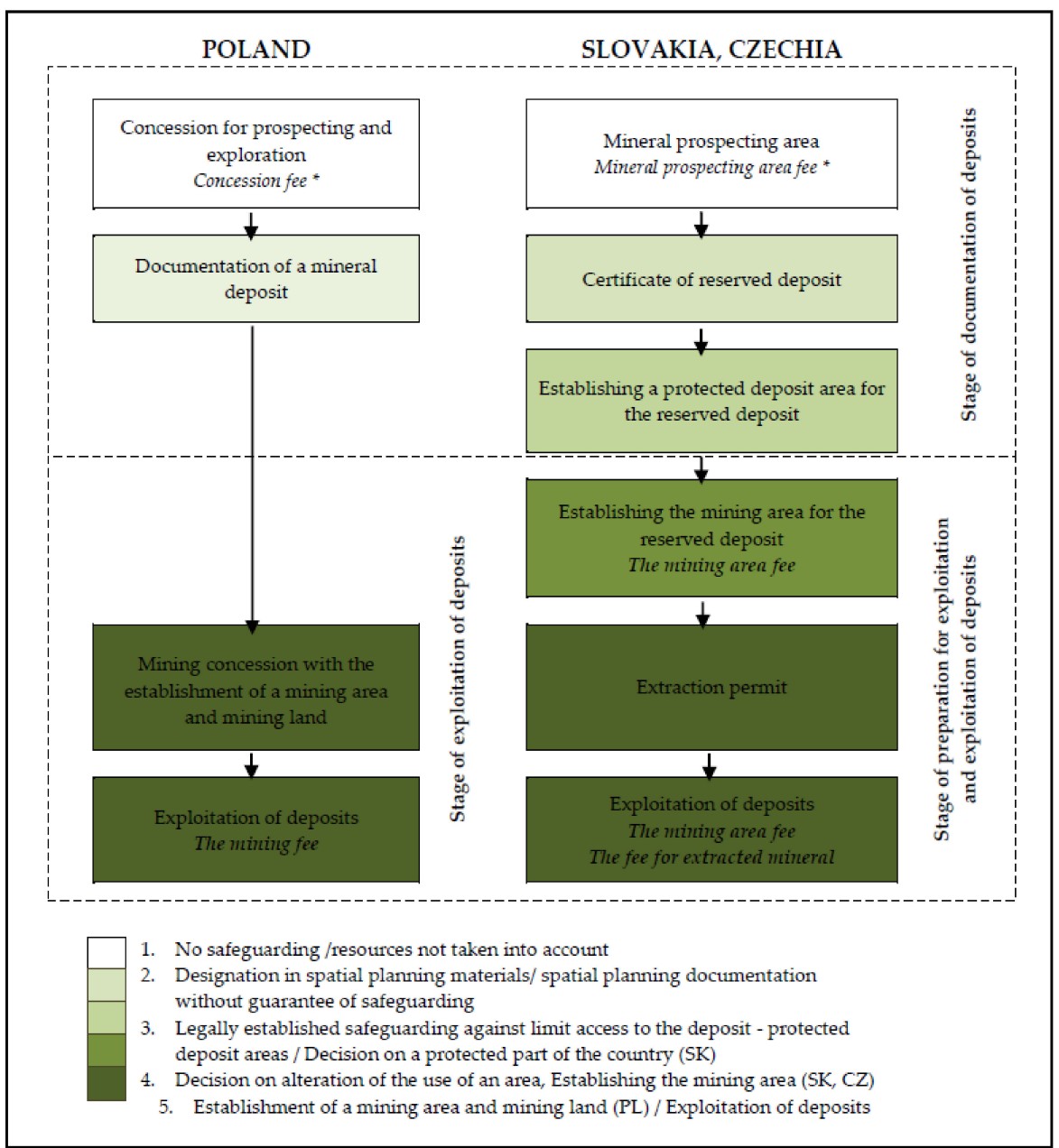

**Figure 3.** Legal, spatial planning and economic instruments in the process of deposit development in Poland, Slovakia and Czechia, *stage of prospecting and related fees are not subject to the analysis (based on: [54–56,96,97]) [own study].

In all analysed countries, establishing the mining area must be deemed a vital deposit safeguarding tool which is supposed to guarantee (apart from environmental safety) the possibility of rational extraction of a mineral, through proper management of the surface above the deposit and in its direct vicinity, which is one of the desired directions of mineral deposit safeguarding. In Slovakia and in Czechia, establishing the MA pertains only to reserved deposits and is not tantamount to the commencement of deposit exploitation, which makes it clearly different to the mining area established in Poland in the process of obtaining concession for mineral extraction. In Slovakia and in Czechia only granting an extraction permit enables the commencement of extraction works. It should be highlighted that obtaining the said permit requires an application submitted by an entrepreneur with the required schedules, such as an agreement on dissolving the conflicts of interests, if not

drawn up at the stage of deposit documentation and delimitation of a protected area for this deposit [55,56]. Through this procedure, the probability of social conflicts at the moment of attempting the deposit extraction is lessened. It should be noted that Polish legislation does not define a similar instrument used for solving conflicts which occur between a mining entrepreneur and land users. In Poland, only in the process of environmental impact assessment (EIA) for selected mining enterprises a comprehensive EIA procedure with participation of the public is envisaged (the EIA procedure is also mandatory in SK and CZ).

In addition, in Slovakia and in Czechia, pointing at the need mineral deposit safeguarding in the cadastral law [101,102] may be considered an element supporting their safeguarding. Land register data are one of the data taken into account in the process of drawing up planning documentations. This aspect was deemed to be sufficiently implemented in safeguarding of deposits, but less effective in their safeguarding (Table 7) due to the most important position of geological surveys [69,103] in the process of data sharing. In Poland the safeguarding of mineral deposits is not a subject of The Geodetic and Cartographic Law [117]. At the same time significant arrangements are presents in The Environmental Protection Law [89] in which minerals are considered as one of the element of the natural environment (Table 7). Due to the above should be protected on the basis of separate regulations.

The stipulations of the EIA Acts [105–107] should also be assessed positively. It should be stressed that a properly conducted procedure for issuing the decision on environmental conditions of approval to a project may also contribute to safeguarding of mineral resources as one of the elements of the environment which may be affected by the investment. A proper impact assessment will help prevent limitations in the access to these resources. This elaboration applies to such investments as construction of roads, an airport or water reservoirs. A similar situation occurs in the case of the strategic environmental impact assessment of a project, e.g., a local spatial development plan, but also (for example) the national road construction programme. Unfortunately, the issue of deposit safeguarding in the broadly understood issue of environmental impact assessment in Poland must be considered underestimated and rarely discussed. Therefore, the effectiveness of this instrument was considered poor (Table 7). In principle, the burden of this safeguarding lies with the authors of reports and forecasts. There is no obligation to notify the Geological Survey of a possible conflicting nature of a given investment. However, an opinion must be issued by a number of other institutions responsible for other components of the environment, for example: The State Water Holding 'Polish Waters' (interference in uniform water bodies), General/Regional Directorate for Environmental Protection (impact on forms of nature and landscape conservation), State Forests (in the case of forest felling) or monument conservation officer (impact on monuments entered in the register of monument protection). It seems that such information would allow to determine the degree of conflict risk and, in some cases, the value of lost benefits for the region or state.

### 4.2.3. Instruments of Spatial Planning

In compliance with the planning instruments of deposit safeguarding, entities responsible for drawing up planning documentations at the local level, on the basis of which administrative decisions are issued, are bound to include in them the information on mineral deposits (PL) and on reserved deposits (SK, CZ) on the basis of available deposit documentations. As a result, in Poland, the planning documentation at the commune level must take into account the conditions related to the occurrence of mineral deposits (developed and undeveloped) as well as mining areas and mining lands delimited for exploited deposits. Moreover, the conversion of land for mining purposes in the planning documentation of a commune is a prerequisite for obtaining a mining concession. Otherwise, a procedure of conversion of land use directions must be sought.

In Slovakia and in Czechia, planning documents are supposed to include boundaries of a reserved deposit, boundaries of a protected deposit area as well as boundaries of

a mining area delimited for the deposit. Such obligation is coming from the Mining law [55,56]. In the Czech Republic, also non-reserved deposits and perspective areas equally as reserved deposits are supposed to be drafted in planning documents, however, it is based on the implementing decree of the Building Act [111]. All the above arrangements do not apply to non-reserved mineral deposits. It constitutes a significant difference as compared to Poland where all documented deposits (i.e., both covered by the mineral rights and the land ownership rights) are treated in the same manner and their occurrence should be incorporated to the planning documentations. Additionally, the possibility and/or necessity (depending on the applicable legal regulations) to obtain opinions, including those concerning deposit safeguarding, should be indicated as a very important element during the creation and updating of all spatial planning instruments. The obligation to obtain an opinion is a very effective tool which can be used to promote interest in mineral deposit safeguarding already at the stage of spatial planning.

The requirement to identify the occurrence of forecasted areas in the studies for planning documentations of a region and a village unit in Slovakia and in Czechia deserves the positive assessment. In Czechia, additionally, the requirement concerns also planning documentations (Table 7). At the same time, this is an element that definitely distinguishes these two countries from Poland, which lacks legal provisions in this scope. Polish legislation does not provide for including deposit forecasted areas in planning documentations of any level. This issue is also not solved in legal instruments for the safeguarding of deposits. It is worth stressing, however, that such a proposal is mentioned in the National Spatial Planning Concept [86], which requires that the planning documentations of each level include potential concession areas for individual minerals. They should be delimited on the basis of exploitation plans for these minerals (also required by the NSDC). Unfortunately, for many years, no steps have been taken toward the implementation of these potential instruments for deposit safeguarding, which are, in principle, right.

### 4.2.4. Economic Instruments

The specific nature of mining activity, consisting in the entrepreneur benefiting from resources of raw materials which are a common good, results in the fact that in mining—both domestic and foreign—in addition to generally applicable taxes and fees, other payments and charges are additionally required. The law regulating mining activities in the analysed countries introduces the so-called mining fee/fee for extracted mineral, and in Slovakia and in Czechia also a fee for the mining area (Figure 3, Table 8).

**Table 8.** Comparison of economic instruments for deposit safeguarding in Poland (PL), Slovakia (SK) and Czechia (CZ); (based on: [54,112,113]). Explanation: assessment of the effectiveness of deposit safeguarding in the scale 1–3, 1—unsatisfactory or below satisfactory, 2—satisfactory, 3—effective or very effective [own study].

| Economic Instruments | Fee Beneficiaries | Assessment of the Effectiveness of Deposit Safeguarding | | |
|---|---|---|---|---|
| | | PL | SK | CZ |
| The mining area fee (SK, CZ) | SK: 20% the state budget, 80% the village budget<br>CZ: 100% the village budget | - | 2 | 3 |
| The mining fee (PL)/The fee for extracted mineral (SK, CZ) | PL *: 60% the commune budget, 40% the budget of Provincial Fund for Environmental Protection and Water Management<br>SK: 100% the budget of environmental funds<br>CZ **: 38% the village budget,<br>62% the state budget | 2 | 3 | 2 |

\* for minerals whose deposits are covered by mineral rights, excluding hydrocarbons, ** depends on the type of mineral, details explained in the text.

The dominant function of the mining fee is the fiscal function and, additionally, the incentive or motivating function [71,118]. Its incentive effect can be seen in the reduction of

its rate in the case of an accompanying mineral and the co-occurring mineral extracted from hydrocarbon deposit [21] or in an increase in the case of activities beyond the provisions of the concession regulating the rules of rational deposit management. In Slovakia, the amount of the fee is affected by, among other things, costs incurred by an entrepreneur for conducting mining activity and costs of reducing or minimising mining damage. In Slovakia and in Czechia, this fee is charged only for reserved deposits (i.e., those owned by the State) and for non-reserved deposits which were considered significant for economic development according to the regulations in force until 1991 and the MA has been delimited for them until 2001 [55,56]. In Poland, it is collected for all minerals. The algorithm of its calculation differs from country to country, but it does not affect its essence. The effectiveness of the mining fee as the economic instrument of deposit safeguarding was assessed as moderate for PL and CZ and good for SK (Table 8).

The mining area fee is understood in Slovakia and in Czechia as a fee paid by an entrepreneur to the state for providing safeguarding of a reserved deposit and granting the right to extract minerals [119]. The purpose of the fee is to motivate the entrepreneur to keep the extraction area as small as possible and to limit the unjustified prolongation of extraction, which meets the demands of rational deposit management and thus its safeguarding. Taking the above-mentioned aspects into account, the effectiveness of the mining area fee in terms of deposit safeguarding was assessed in Slovakia as moderate and in Czechia as good (Table 8). Unfortunately, determining the obligation to pay the mining are fee partly supports speculation in mining areas. For example, one company may have more mining areas (three deposits–three MAs), but it extracts only one of them, blocking the other ones from possible competition. In this manner, raw material prices can influence and increase the demand for raw materials covered by imports.

The undoubted disadvantage of fees related to mineral deposits used in the analysed countries as an economic instrument for deposit safeguarding can be the lack of purpose in the case of communes (PL) and village units (SR, CZ) in areas where minerals are extracted. Important beneficiaries of the above-mentioned fees (Table 8) are local governments, which, however, may use the received funds as they see fit. Only funds from the state budget or the budget of environmental funds are earmarked for the elimination of mining damage caused by the extraction of deposits, for securing and eliminating abandoned mining works, or for land reclamation and revitalisation as well as for ensuring that the activities of the state geological survey are carried out, and are a source of covering the so-called long-term obligations in the area of geology and mining [54–56,89].

## 5. Conclusions

In Poland, Slovakia and Czechia the deposits safeguarding is a subject to formal-legal conditions, both in terms of safeguarding of access to deposits and proper (rational) management of deposit resources at the exploitation stage. Ensuring of access to deposit is a topic of land use policy (by appropriate legal acts regarding spatial planning) and also mining law (e.g., obligation to determine of the PDA in Slovakia and Czechia and obligation to disclose all documented deposits in the local spatial documents in Poland). Safeguarding of deposits, understood as the rational and comprehensive use of minerals, is the subject of mining law only. All countries have a documented resource base, and designated deposits have specific boundaries and calculated resources. The vast majority of them have never been exploited. However, a potential mining entrepreneur has the right to use geological information of deposits which is facilitated by the national geological surveys.

The countries analysed in the paper are characterised by an extremely different resource base, both in terms of quantity and type of minerals. However, regardless of the number of documented deposits and the size of their resources, they should in any case be subject to comprehensive safeguarding in line with principles of sustainable development. A comprehensive safeguarding of deposits in a given country should concern both forecasted areas of mineral occurrence as well as documented deposits, in operation or not. Unfortunately in the analysed countries safeguarding of forecasted areas is only slightly

emphasised. In Poland they are a subject of the mineral policy (as a conceptual instrument) and the National Spatial Planning Concept (as one of spatial planning instruments). It is not developed further in the legal and economic instruments. They are not included in the mining law and the law responsible for domestic spatial planning policy. It should be concluded clearly that in Poland safeguarding of forecasted areas does not exist. In Slovakia and Czechia the occurrence of forecasted areas is considered in spatial planning documents, but still without any guarantee of safeguarding.

Among the instruments of deposit safeguarding analysed, conceptual instruments play an overarching role, the most important of which is the properly formulated country's mineral policy that clearly defines directions of activities towards deposit safeguarding. These, in turn, should be developed and detailed in the relevant legislative acts, for example in the field of mining and geology and spatial planning.

The assessment of mineral policies of the countries discussed in this paper is the least favourable in the case of Poland that, as a matter of fact, does not have such a document. Although the authors of this paper analyse its draft version which, in its objectives, is right and comprehensively implements deposit safeguarding, implementation of measures cannot be discussed when, formally, such a document does not exist. Perhaps this is one of the reasons why other instruments indicated in this paper are in Poland characterised in lower effectiveness of safeguarding when compared to Slovakia and the Czech Republic. Therefore, it seems necessary to press in Poland on further work on the adoption of the Mineral Policy, and then on the implementation of its provisions, and in the other countries analysed ensuring that the arrangements of this document are periodically updated.

In Slovakia and in Czechia, legal safeguarding of a reserved deposit is achieved by establishing a PDA and a MA for this deposit. This scope and method of mineral deposit safeguarding, supported additionally by spatial planning instruments and PDA registration in the cadastral system can be considered satisfactory. Among the legal and planning instruments for deposit safeguarding in Poland, it is difficult to find similar and equally effective solutions. The obligation to disclose documented deposits in planning documentations does not provide sufficient grounds for real deposit safeguarding, despite the existence of general provisions indicating its need. The existing forms and methods of mineral deposit safeguarding in spatial planning in Poland are not satisfactory to resolve conflicts of interest concerning the management of the area above the deposit, the management of co-occurring environmental resources subject to protection and securing the possibility of deposit exploitation, regardless of the degree of their identification.

Rightly conducted spatial planning is undoubtedly responsible for appropriate and efficient deposit safeguarding. However, the guidelines on the methods and rules of protection and its implementation in spatial planning are already included in the relevant mining laws. It is regarding Poland where mining law obligate local authorities to take into account all the deposits in local spatial documents. Similarly in Slovakia and Czechia mining law determines the protection deposit areas for selected deposits as a key instrument of deposit safeguarding. To sum up, in the analysed countries the safeguarding of deposits is defined in the mining laws and is implemented through spatial planning instruments.

In reference to the mining law in Slovakia and in Czechia, a request about the necessity of updating it and applying a newer approach to the analysed issue needs to be made. The binding law, in fact, originates in the times of the Socialistic Republic of Czechoslovakia. Currently, the law should be based on individual economic, infrastructural and technological conditions as well as on the need of economic growth in the individual regions of those countries. In Slovakia and in Czechia, current circumstances pertaining to non- reserved deposits should be discussed as they are not safeguarded under the current legal system. Since the early 1990s, as a result of the amendment of the Mining Law [120], the recognition of non-reserved deposits of a commercial nature as reserved deposits was abandoned [121]. Moreover, the importance of non-reserved deposits, especially in case of sand and gravel is growing [75]. These are mainly building materials, which are necessary for economic development of the regions where they occur. So far, the resources of both countries in

terms of these raw materials can be considered sufficient, however, in many cases the exploited deposits are depleted and new ones (although with good quality raw material) are difficult to open. The actual availability of these raw materials will therefore depend, in future, on how the mining sector is able to cope, apart from administrative complications, with issues of public access and with different groups of interest in the potential extraction of a mineral.

The core of mining fees as economic instruments of deposit safeguarding is similar in analysed countries. However, the key importance of mining area fees in Slovakia and Czechia should be pointed out. The dominant motivation function of them is relevant in rational management of deposit, which is one of the mineral safeguarding directions.

To sum up, it should be pointed out that the natural uniqueness of mineral deposits deserves more attention when safeguarding and using them. Therefore, there is a constant need to look for new solutions and good practice in safeguarding the availability of their resources. When considering these issues, it is also important to bear in mind the significant economic issues in both local, national and global dimensions. Insufficient or lacking safeguarding of domestic mineral deposits leads to the dependence of a given economy on raw material supplies. Social awareness should also be at the heart of effective deposit safeguarding. It is necessary to create a realistic picture of the need for mining in the society, instead of just highlighting its negatives. The current European generation has low awareness of the importance of mineral resources, despite their high consumption.

In the future, we should look at mineral deposits as a natural stimulus and a unique natural opportunity to develop new technologies in production, processing and use in society, and as opportunities for regional development, rather than ecological or environmental loads.

**Author Contributions:** Conceptualisation: S.G., A.K.-N.; methodology: S.G., A.K.-N., A.G.; software: S.G., A.K.-N.; validation: S.G., A.K.-N.; formal analysis: S.G., A.K.-N., A.G., B.W., J.K.; investigation: S.G., A.K.-N., A.G., B.W., J.K.; resources: S.G., A.K.-N.; data curation: S.G., A.K.-N.; writing—original draft preparation: S.G., A.K.-N., A.G.; writing—review and editing: S.G., A.K.-N., A.G., B.W., J.K.; visualisation: S.G., A.K.-N.; supervision: S.G., A.K.-N.; project administration: S.G; A.K.-N.; funding acquisition: A.K.-N. All authors have read and agreed to the published version of the manuscript.

**Funding:** This article has been supported by the Polish National Agency for Academic Exchange under Grant No PPI/APM/2019/1/00079/U/001.

**Institutional Review Board Statement:** The study did not involve humans or animals.

**Informed Consent Statement:** The study did not involve humans.

**Data Availability Statement:** The study did not report any data.

**Acknowledgments:** This research was written as part of research in projects by the Polish National Agency for Academic Exchange under Grant No PPI/APM/2019/1/00079/U/001, AGH University of Science and Technology, Faculty of Geology, Geophysics and Environment Protection: 16.16.140.315 and VEGA 1/0585/20.

**Conflicts of Interest:** The authors declare no conflict of interest.

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
