# Peer review of "Instruments of Mineral Deposit Safeguarding in Poland, Slovakia and Czechia—Comparative Analysis"

_resources, doi:10.3390/resources10020016_

Round 1

Reviewer 1 Report

I recommend tо the authors reveal the scientific problem in the introduction of this paper. Also more clearly formulate the aim and objectives of the study.

Author Response

  1. I recommend tо the authors reveal the scientific problem in the introduction of this paper. Also more clearly formulate the aim and objectives of the study.

Required elements was clarified in the Introduction chapter – lines 124-131: "There is, however, a constant need of searching for innovative, more efficient tools adopted to current and future circumstances. The aim of this paper is conducting  the multidimensional assessment of instruments used in Poland, Slovakia and Czechia for safeguarding forecasted areas of mineral occurrence and their documented deposits. This is the first scientific paper considering above issue that was developed in an international consortium. At the same time, identification of the best, most effective examples of management instruments for rational use and safeguarding of deposits is the essential subject of the study."

Reviewer 2 Report

The introduction is appropriatein terms of its content and addressing the topic, but it is somewhat wordy. I would recommend the authors to shorten it: a 20-25% less would really enhance its readability. I have found the same problem throughout the whole manuscript. It is not that it is unduly organized, but a more profuse sub-indexing would allow the reader to better locate him/herself within the train of though in this manuscript.

It may not be the subject of this study, but I think that the conceptual framework of the ore deposits should be more specific on geological (i.e. models for the deposits) and mining aspects, whether it is legally required or not. Such are the basis of the environmental assessment of such a heritage and whatever legislation may be cast on the matter.

Author Response

  1. The introduction is appropriate in terms of its content and addressing the topic, but it is somewhat wordy. I would recommend the authors to shorten it: a 20-25% less would really enhance its readability. I have found the same problem throughout the whole manuscript. It is not that it is unduly organized, but a more profuse sub-indexing would allow the reader to better locate him/herself within the train of though in this manuscript. The Introduction was partly shortened (lines 56-61, 66-75, 81-84, 114-117), the structure of the article was changed (especially chapters Results, Discussion and Conclusions)
  2. It may not be the subject of this study, but I think that the conceptual framework of the ore deposits should be more specific on geological (i.e. models for the deposits) and mining aspects, whether it is legally required or not. Such are the basis of the environmental assessment of such a heritage and whatever legislation may be cast on the matter. - Due to the comprehensive subject matter covered in the article, it was not decided to further consideration the safeguarding of the deposits depending on their geological specificity.

Reviewer 3 Report

First of all, the manuscript is very interesting, however it needs some improvements. Detailed comments are made in the attached file. Here I make general comments.

The first one is the need to clarify what you mean by "mineral deposit safeguarding". Following the UK concept, safeguarding minerals is the Land-Use Planning act of ensuring that areas containing, or potentially containing, mineral resources are not needlessly occupied by other uses that may prevent their future extraction. It means that, if needed, those areas may be designed to other uses, even in the presence of a mineral deposit - It is a soft policy. 

This concept is different from the one you are using, which, in my opinion, relates to the strict protection of the area where minerals exist (e.g. the SL and CZ Protected Deposits Area) - A hard policy tool.

Therefore, you have to clearly identify what you are talking about. If you are talking about both, then you should do it in, let's say, different sections.

My second main comment: The Results chapter needs to be better structured. Presenting much of the information in the form of tables may help.

Third: The same applies for the Discussion chapter. Many paragraphs should be in the Results. The text is to dense to be easly understandable. See my detailed comments.

Fourth: The conclusions should be improved: see detailed comments

Finally: Text needs english improvements. I made minor suggestions.

Author Response

General comments to review number 3:

- definition of mineral deposit safeguarding was introduced in „Introduction” chapter,

- the structure of „Results”, „Discussion” and „Conclusions” was changed,

- detailed comments and answers are listed below.

Detailed comments and answers:

  1. [Line 21] “mineral deposits can be found is based on the statement that they can be extract only in places where they occur” - corrected - lines 22-23
  2. [Line 29] “are” - corrected - line 30
  3. [Line 31] “effectiveness” - corrected - line 32
  4. [Line 33] “Legal instruments” - not included, because "instruments" generally applies to all assessed instruments, not only legal, but also conceptual and economic instruments
  5. [Line 41] “Emissions” - corrected - line 42
  6. [Line 43] “Needs English improvement. Maybe: The environment is a set of natural elements, including those transformed as a result of human activity, in particular the following:” - corrected - line 45-46
  7. [Line 74] “... successfully. However, a constantly ...” - corrected - line 79
  8. [Line 76] “Simplify. Maybe: The most important ones are the development of urban, agricultural and tourism areas, the latter requiring high environmental quality standards, being landscape part of it.” -corrected - lines 84-86
  9. [Line 81] „No need for this. The whole idea could be like this: And in accordance with ....degradation of the environment, forgetting that current requirements for the mines in respect ... stringent. Therefore, ....” - corrected - lines 87-88
  10. [Line 84] “You haven't talked about "depletion" yet. So, bringing the "depletion issue" to here complicates the main idea, which is: The prymary issue for a successfully rationale and sustainable supply of minerals to society is granting the access to them in spatial planning. You should use: Mateus et al., 2017 (https://doi.org/10.1007/s13563-017-0114-y)” - the sentence was corrected – line 92; the proposed article has been analysed and included in the text – line 96
  11. [Line 88] „Taking into account your biblio reference no. 19, as well as Wrighton, C.E., et al., 2014 (Resources Policy 41,), you should clarify here what this concept means. Attention: it does not mean "protection of mineral deposits", but "protection of access to them". Therefore, for instance in the next phrase "... instruments of its safeguarding...", I am understanding "instruments to protect the access to minerals in LUP", not "instruments to protect mineral deposits" as it is the case of a mining concession. I will highlight without commenting the phrases the meaning should be clarified.

definition of mineral deposit safeguarding was introduce in Introduction chapter – lines 96-103

According to Authors – and taking into account the formal-legal conditions of analysed countries – mineral deposits safeguarding is a multidimensional topic and includes following elements:

  1. safeguarding of land access and safeguarding of area against development, which may prevent the use of the deposit and the necessary mining activities,
  2. resources safeguarding against unjustified losses and minimizing unavoidable losses (also resulting from inappropriate land development),
  3. complete use of documented resources of deposit including accompanying mineral.

Thus, the deposit safeguarding includes both the safeguarding to access to them and their rational and efficient management during mining operations. In this sense, it is analysed in present article.

  1. [Line 93] highlighted text – explanation in the text (lines 96-103) and in the answer to comment 11
  2. [Line 102] highlighted text – explanation in the text (lines 96-103) and in the answer to comment 11
  3. [Line 124] “To clarify my last comment, I also make a comment here: Which instruments? For protect/ensure the access to mineral deposits im land use planning? (i.e. Minerals Safeguarding concept) or for protect the mineral deposits themselves (e.g. mining concession)? or both?” both - explanation in the text (lines 96-103) and in the answer to comment 11
  4. [Line 159] highlighted text – explanation in the text (lines 96-103) and in the answer to comment 11
  5. [Line 162] “Biblio references” - completed in text – line 18: Nevertheless, there have been numerous discussions on whether the mineral deposit policy at the European level should exist and, if so, which issues should it address within the framework of the European sustainable industrial strategy [20,43,45].
  6. [Line 168] „taken equally” - corrected - line 189
  7. [Line 177] „each country is supposed to identify and safeguard access to significant mineral deposits in accordance with its national legislation” - corrected – lines 201-202
  8. [Line 241] “will be” - corrected – line 264
  9. [Lines 251-252] „the reserved mineral deposits form a mineral wealth property of the state.” - corrected – lines 275-276
  10. [Line 254] „You can resume all this because it is the same for the 3 countries: State owned mineral deposits and private owned mineral deposits (belonging to the owner of the land).” -The authors decided to leave the text unchanged. Polish legislation covers the protection of all deposits (belonging to the State Treasury and the landowners). In the Czechia and Slovakia, only reserved deposits are legally protected. This is an important difference that the Authors wanted to emphasize at this point.
  11. [Table 1] „not mineral deposits” - explanation was added in table 1 (NOT minerals - natural resources that are not defined as mineral raw materials and for which there are no documented deposits)
  12. [Line 285] „???” - removed – line 361, additional explanation for Reviewer – MIDAS it is an acronym of the Polish (mineral) geological database
  13. [Line 384] highlighted text – explanation in the text (lines 96-103) and in the answer to comment 11
  14. [Line 524] “A comment just to frame the discussion: This scheme illustrates that minerals safeguarding was previously achieved. I mean, if the deposit is documented it is because access to land was allowed previously (mining companies hardly will carry exploration works in areas where they know that exploitation will be interdicted). If the are where the deposit occurs will be i) secured for further exploration or even exploitation, or ii) if it will ever be exploited or not, are different issues. In the case i) it is a decision taken at LUP dependent on weighing other options. In the case ii) depends on EIA”
    • Explanation for Reviewer: In the analyzed countries, documenting the deposit is not synonymous with starting its exploitation. Even if the mining entrepreneur decides to document the deposit, the environmental impact assessment (EIA) procedure may prevent this exploitation. It is relevant element of investment risk. Moreover, expanded and documented resources bases of each countries are (in large part) result of carrying tasks of domestic geological surveys (mainly in the past). Documenting the deposits was aimed at recognizing the resource potential of countries. Most of the deposits documented in this way have defined boundaries, calculated resources and detailed geological documentation, but they have never been object to mining exploitation.
  15. [Line 572] highlighted text – explanation in the text (lines 96-103) and in the answer to comment 11
  16. [Line 578] highlighted text – explanation in the text (lines 96-103) and in the answer to comment 11
  17. [Line 762] highlighted text – explanation in the text (lines 96-103) and in the answer to comment 11
  18. [Line 767] “You are repeating (at least partially) what was already presented. Moreover, these are results, not discussion. So, I would say something like: As can be seen by the comparison presented in table 4, ....(highlight what you do want to compare..)” corrected - part of the text (about mineral potential of countries) together with table 4 (now table 3) was moved to Results chapter – lines 335-354 and table 3
  19. [Line 785] “However, all this relates to table 1, not to table 4. Therefore, what is the meaning of table 4?” corrected – see comment 29, table 4 (now table 3) is connected with table 1 - Mineral deposits classification applied in Poland, Slovakia and Czechia. Table 3 relates to mine production of minerals from deposits covered by the mineral right (PL)/reserved deposits (SK, CZ)/ and deposits covered by the ownership right to the land property (PL)/deposits of non-reserved minerals (SK, CZ)
  20. [Line 809] “All this chapter is very difficult to read. It is information, after information, and so on, especially after line 28 of page 23. Moreover, it presents information that was not presented in results. Perhaps, a good solution is to present a subchapter in the results section describing the permitting procedures for each country and respective main constraints. Then, the discussion could be made in a more resumed and understandable way. In addition, the discussion of elements from table 6 should be cleary sectioned according to Mining Act, environmental law, etc.”

All comments of the Reviewer were taken into account:

  • The entire chapter has been redrafted - sub-chapters have been supplemented, the correct order of the text has been corrected, which significantly improved the readability of the chapter – lines 902, 1, 77, 1
  • The issue of " permitting procedures" was not the subject of the work, it is a very extensive topic, perhaps for a separate manuscript, only the most important differences related to instruments of mineral deposit safeguarding, were discussed in the manuscript
  • Paragraphs introduced so that the results of table 6 (now in table 7) are correctly and legibly discussed.
  1. [Line 811] “I don't understand? the analysis must be proceded by defining the authorities responsible by the analysis???? – corrected- line 894: A detailed comparative analysis of identified instruments of mineral deposit safeguarding in Poland, Slovakia and Czechia needs to be proceeded by defining authorities at the national administration level competent in the field of deposit safeguarding.
  2. [Line 841] “Before you presented some constraints governing the comparison. Now you start to compare. Therefore, this must be a new paragraph.” - corrected – lines: 921, 927
  3. [Line 845] “because ....” changed - line 928
  4. [Line 846] “because ...You should justify the classification whenever you use it” - The authors do not introduce additional explanation, the assessment is made from the text above, it is detailed in the paragraphs: 904-927
  5. [Table 5] “Good table. However, I'm again affraid about the meaning of "deposits safeguarding". Is it safeguarding the access or safeguarding the deposit? Usually, safeguarding the access is a matter of land use policy, protecting the deposit is a matter of mining policy
    • The Authors agree with the Reviewer's opinion that protection of access (to deposits) is generally the subject of spatial policy, and the protection of exploited deposits is subject of mining law (policy). Nevertheless - as supplemented in the introduction - the safeguarding of deposits covers both the protection of access to them and the rational management of resources during the exploitation.
  6. [Line 2] highlighted text – explanation in the text (lines 96-103) and in the answer to comment 11
  7. [Line 7] “It is repeated below” – the text is not repeated, in this paragraph legal instruments are assessed (table 6 (now table 7)), the above text related to conceptual instruments
  8. [Line 8] “So, it is an old law, that needs to be updated, but it is satisfactory regarding safeguarding! Are the considerations about updating the law needed here?” – yes, the law is satisfactory regarding safeguarding, but the considerations about updating the law are needed – lines 11-12 and 67-77; this part of the Discussion was moved to Conclusions (in accordance with comment 56)
  9. [Line 13] highlighted text - explanation in the text (lines 96-103) and in the answer to comment 11
  10. [Line 20] “for other purposes than mining!” - corrected – line 22
  11. [Line 22] “Not relevant” – The authors believe that this is important, above all in terms of spatial planning and social conflicts.
  12. [Line 27] “... different resource base, both ...” - corrected – line 67
  13. [Line 35] „to consider a comprehensive safeguarding ...” - corrected – line 75
  14. [Line 36] „I was not able to read (or understand) in the discussion how the forecast areas are taken into account” - the manuscript considers the issue of safeguarding forecast areas in several places:
  • 2. Instruments of mineral deposit safeguarding, 3.2.2. Legal instruments, lines 604-607: „According to the mining law in Slovakia and Czechia, the spatial planning authorities and authorities drawing up spatial planning documentation are obliged to conduct their activities based on the documentation of probable deposit areas and reserved mineral deposits provided to them by the Ministry of Environment”; probable corrected to forecasted:
  • 2. Instruments of mineral deposit safeguarding, 3.2.3. Instruments of spatial planning, lines 765- 769: „These documents are relevant to safeguarding of mineral resources as they assess the condition and development of an area, determining the so-called limits of land use, i.e. restrictions in land use due to the protection of general interests, including mineral deposits (forecasted reserved deposits, reserved deposits and non reserved deposits), protected deposit areas and mining areas”.
  • Discussion of the results, 4.2.1. Conceptual instruments: lines 917-918: „In Poland, the draft of mineral policy refers also to forecasted areas what is missing in Slovakia and in Czechia.”,
  • Discussion of the results, 4.2.3. Instruments of spatial planning, lines 112-123: „The requirement to identify the occurrence of forecasted areas in the studies for planning documentations of a region and a village unit in Slovakia and in Czechia deserves the positive assessment. In Czechia, additionally, the requirement concerns also planning documentations”, „Polish legislation does not provide for including deposit forecasted areas in planning documentations of any level. This issue is also not solved in legal instruments for the safeguarding of deposits. It is worth stressing, however, that such a proposal is mentioned in the National Spatial Planning Concept [86], which requires that the planning documentations of each level include potential concession areas for individual minerals.”
  • Table 5 (now table 6): Safeguarding of forecasted areas (Y/N)
  • Table 6 (now table 7): Safeguarding of forecasted areas (Y/N)
  • Terminology was standardized
  1. [Line 36] „as well as documented deposits, in operation or not.” - corrected – line 77
  2. [Line 41] „... detailed in the relevant related legislative acts. Any other conclusions about Conceptual instruments besides having an overarching role?” - supplemented about legal instrument, instrument of spatial planning and economic instrument – lines 57-66, 78-85, 120-147 (related to the comment 56)
  3. [Line 43] „"Regarding mineral policies, the least favourable is the draft version of Poland." You do not need to repeat again that it was not yet implemented, etc., etc,” – the authors decided to leave this information for potential readers who would like to read only conclusions
  4. [Line 47] „Not needed to talk about who is rich and poor.” – deleted- lines 95-96
  5. [Line 54] „This is for the Introduction section” deleted, changed – lines 103-106
  6. [Line 57] “I don't understand. Spatial planning instruments are legal instruments! So, why to refer that they show a strong link to ...” - deleted – lines 106-107
  7. [Line 58] “I agree, but from the discussion I was not able to understand it.” - deleted – lines 107-108, changed – lines 120-127: Rightly conducted spatial planning is undoubtedly responsible for appropriate and efficient deposit safeguarding. However, the guidelines on the methods and rules of protection and its implementation in spatial planning are already included in the relevant mining laws. It's regarding Poland where mining law obligate local authorities to take into account all deposits in local spatial documents. Similarly in Slovakia and Czechia where mining law determine  the protection deposit areas for selected deposits as a key instrument of deposit safeguarding. To sum up, in the analysed countries the safeguarding of deposits is defined in the mining laws and is implemented through spatial planning instruments.
  8. [Line 63] “Among the ...” - corrected, line: 112
  9. [Line 67] “As this is written for Poland in contrast to Sl and Cz, can we conclude that PDA resolve all this issues?” - supplemented – lines 134-143
  10. [Line 71] “These are not conclusions, but rather personal statements. Some of them can be included in the introductory text (supported by biblio refs) for justifiyng the need of safeguarding mineral deposits.” part of text was removed - lines: 153-159, 163, 166-168; not included in the introductory text (Reviewer 1 suggest to shorten the Introduction)
  11. [Line 91] “I believe that from the dense Discussion chapter you could retrieve some other conclusions organized by each type of legislative documents. And then, a general conclusion about the effectiveness of the whole system in each country. As it is now, this Conclusions chapter seems only like a way to justify the need of a minerals policy in Poland.” the structure and contents of Conclusions was changed (related to the comments 47, 52, 54), part of text was removed (related to the comments 50, 51, 52, 55).

Round 2

Reviewer 3 Report

I accept your corrections and explanations